# Rational design of highly potent broad-spectrum enterovirus inhibitors targeting the nonstructural protein 2C

Lisa Bauer[1], Roberto Manganaro[2], Birgit Zonsics[2], Daniel L. Hurdiss[1], Marleen Zwaagstra[1], Tim Donselaar[1], Naemi G. E. Welter[3], Regina G. D. M. van Kleef[3], Moira Lorenzo Lopez[2], Federica Bevilacqua[2], Thamidur Raman[2], Salvatore Ferla[2], Marcella Bassetto[4], Johan Neyts[5], Jeroen R. P. M. Strating[1¤], Remco H. S. Westerink[3], Andrea Brancale[2], Frank J. M. van Kuppeveld[1]*

1 Virology Section, Infectious Disease and Immunology Division, Department of Biomolecular Health Sciences, Faculty of Veterinary Medicine, Utrecht University, Utrecht, The Netherlands, 2 Medicinal Chemistry, School of Pharmacy & Pharmaceutical Sciences, Cardiff University, Cardiff, United Kingdom, 3 Neurotoxicology Research Group, Toxicology Division, Institute for Risk Assessment Sciences (IRAS), Faculty of Veterinary Medicine, Utrecht University, Utrecht, The Netherlands, 4 Department of Chemistry, Swansea University, Swansea, United Kingdom, 5 Department of Microbiology, Immunology and Transplantation, Rega Institute for Medical Research, KU Leuven, Leuven, Belgium

¤ Current address: Viroclinics, Rotterdam, The Netherlands
* f.j.m.vankuppeveld@uu.nl

**Data Availability Statement:** All relevant data are within the paper and its Supporting information files.

## Abstract

There is a great need for antiviral drugs to treat enterovirus (EV) and rhinovirus (RV) infections, which can be severe and occasionally life-threatening. The conserved nonstructural protein 2C, which is an AAA+ ATPase, is a promising target for drug development. Here, we present a structure-activity relationship study of a previously identified compound that targets the 2C protein of EV-A71 and several EV-B species members, but not poliovirus (PV) (EV-C species). This compound is structurally related to the Food and Drug Administration (FDA)-approved drug fluoxetine—which also targets 2C—but has favorable chemical properties. We identified several compounds with increased antiviral potency and broadened activity. Four compounds showed broad-spectrum EV and RV activity and inhibited contemporary strains of emerging EVs of public health concern, including EV-A71, coxsackievirus (CV)-A24v, and EV-D68. Importantly, unlike (S)-fluoxetine, these compounds are no longer neuroactive. By raising resistant EV-A71, CV-B3, and EV-D68 variants against one of these inhibitors, we identified novel 2C resistance mutations. Reverse engineering of these mutations revealed a conserved mechanism of resistance development. Resistant viruses first acquired a mutation in, or adjacent to, the α2 helix of 2C. This mutation disrupted compound binding and provided drug resistance, but this was at the cost of viral fitness. Additional mutations at distantly localized 2C residues were then acquired to increase resistance and/or to compensate for the loss of fitness. Using computational methods to identify solvent accessible tunnels near the α2 helix in the EV-A71 and PV 2C crystal structures, a conserved binding pocket of the inhibitors is proposed.

**Funding:** This work was supported by research grants from The Netherlands Organisation for Scientific Research (NWO-ECHO-711.017.002 to F. J.M.v.K. and J.R.P.M.S.; NWO-VICI-91812628 to F. J.M.v.K.) and the European Union (Horizon 2020 Marie Skłodowska-Curie ETN "ANTIVIRALS", grant agreement number 642434 to J.N, A.B., and F.J.M. v.K.). D.L.H. is funded from the European Union's Horizon 2020 research and innovation program under the Marie Skłodowska-Curie grant agreement (No 842333) and holds an EMBO non-stipendiary long-term Fellowship (ALTF 1172-2018). M.B. and S.F. were supported by the Sêr Cymru programme which is part-funded by Cardiff University and the European Regional Development Fund through the Welsh Government. The funders had no role in study design, data collection and analysis, decision to publish, or preparation of the manuscript.

**Competing interests:** The authors have declared that no competing interests exist.

**Abbreviations:** AFM, acute flaccid myelitis; AFP, acute flaccid paralysis; ATCC, American Type Culture Collection; BGM, Buffalo Green Monkey; Boc, *tert*-butoxycarbonyl; $CC_{50}$, 50% cytotoxic concentration; $CCID_{50}$, 50% cell culture infective dose; CDI, 1'-Carbonyldiimidazole; COPD, chronic obstructive pulmonary disease; CPE, cytopathic effect; CV, coxsackievirus; DAT, dopamine transporter; DIPEA, diisopropylethylamine; DMEM, Dulbecco's Modified Eagle Medium; $EC_{50}$, 50% effective concentration; ECACC, European Cell Culture Collection; EV, enterovirus; FBS, fetal bovine serum; FCS, fetal calf serum; FDA, Food and Drug Administration; GuaHcl, guanidine hydrochloride; HeLa, Henrietta Lacks cell line; HRV, human rhinovirus; $IC_{50}$, 50% inhibitory concentration; IMDM, Iscove's Modified Dulbecco's Medium; IRES, internal ribosomal entry side; MBR, mean burst rate; MEM-NEAA, minimum essential medium non-essential amino acids solution; MNBR, mean network burst rate; MOI, multiplicity of infection; MSR, mean spike rate; mwMEA, multiwell micro-electrode array; NET, norepinephrine transporter; PBS, phosphate-buffered saline; PDB, protein database; PEI, polyethyleneimine; PV, poliovirus; RD, Rhabdomyosarcoma; RV, rhinovirus; SERT, serotonin transporter; SF3, superfamily 3; SSRI, selective serotonin reuptake inhibitor; TBTU, 2-(1*H*-benzotriazole-1-yl)-1,1,3,3-tetramethyluronium tetrafluoroborate; $TCID_{50}$, 50% tissue culture infectious dose; TFA, trifluoroacetic acid; TR, treatment ratio.

## Introduction

The genus *Enterovirus* of the family Picornaviridae is a large group of nonenveloped, positive-sense, single-stranded (+) RNA viruses. Four enterovirus species (EV-A to -D) and three rhinovirus species (RV-A to -C) constitute the set of human pathogens that have large medical and socioeconomical impact, such as poliovirus (PV), coxsackievirus (CV) A and B, echoviruses, numbered EVs (e.g., EV-A71, EV-D68) and RVs (e.g., human rhinovirus [HRV]-A2, HRB-14) [1]. Though often unnoticed and self-limiting, EV infections can cause serious illnesses and be associated with major complications, which can be life-threatening, especially in infants, young children, and immunocompromised individuals. Infections with EVs can cause a broad range of different clinical manifestations, including hand-foot-and-mouth disease, conjunctivitis, aseptic meningitis, myocarditis, severe neonatal sepsis-like diseases, respiratory diseases, acute flaccid paralysis (AFP), and acute flaccid myelitis (AFM). In recent years, several EVs have been considered as pathogens of increasing health concern. These include EV-A71 and EV-D68—of which large outbreaks in South East Asia and the United States, respectively, are associated with severe neurological complications—as well as CV-A24v (an EV-C species member), which causes large pandemics of a highly contagious conjunctivitis [2–5]. RVs are the causative agent of the common cold and can trigger exacerbations of asthma and chronic obstructive pulmonary disease (COPD) [1].

Several strategies may be used to control EV infections. One strategy involves the development of vaccines. Inactivated and life attenuated vaccines have been developed against PV, and recently, inactivated vaccines against EV-A71 were approved in China [6]. However, given the large number of serotypes (>100 nonpolio EVs and >150 HRVs), development of a pan-EV and -RV vaccine seems unfeasible. Another strategy is the development of potent antivirals. Several EV inhibitors have been identified. These include both direct-acting antivirals, most of which bind to the viral capsid or the viral protease 3C, as well as inhibitors that target host factors essential for virus replication (reviewed by Bauer and colleagues [7]). These inhibitors were tested in clinical trials, but their development was halted due to limited efficacy, poor bioavailability, or toxicity issues. At present, no antiviral against EVs is licensed for therapeutic use.

An attractive target for antivirals is the highly conserved and multifunctional nonstructural protein 2C. 2C is an ATPase associated with diverse cellular activities (AAA+ ATPase) classified within the superfamily 3 (SF3) helicases. These enzymes couple the hydrolysis of ATP to movement of protein domains which, in turn, drive the unwinding of a nucleic acid substrate. It has been shown biochemically that 2C functions as RNA helicase and ATP-independent RNA chaperone [8–11]. 2C fulfills pleiotropic functions during the virus life cycle, including replication organelle formation, genome replication, and encapsidation [12–19]. Screening of drug libraries identified many structurally disparate 2C inhibitors such as the Food and Drug Administration (FDA)-approved drugs fluoxetine, dibucaine, pirlindole, and zuclopenthixol [20–23]. One of the most promising candidates is fluoxetine, a selective serotonin reuptake inhibitor (SSRI) that is approved for the treatment of depression and anxiety disorders. Fluoxetine inhibits replication of viruses belonging to the EV-B and EV-D species and some RV, but not of viruses belonging to EV-A and EV-C species [21, 23, 24]. Fluoxetine has been used off-label to successfully treat an immunocompromised child with chronic EV encephalitis [25]. Recently, the safety and efficacy of fluoxetine for treatment of EV-D68 associated AFM was investigated in a retrospective study. The treatment with fluoxetine revealed no clinical benefit but rather suggested a worsening of the patient conditions in the fluoxetine-treated cohort [26]. The reason for this is unknown but might be related to the drug's SSRI activity. We previously established that fluoxetine inhibits viral replication stereospecifically by directly binding

2C [24]. Unfortunately, the chemical moiety important for the SSRI activity is essential for the antiviral activity, and thus far these two activities could not be uncoupled [27]. This raises concerns about the therapeutic application of fluoxetine and shows that other, potent, biosafe, and broad-spectrum antiviral inhibitors are needed.

In a high-throughput screen of small molecules, the compound N-(4-fluorobenzyl)-N-(4-methoxyphenyl)furan-2-carboxamide (which will further be referred to as compound 1) was identified as a potential CV-B3 2C inhibitor [20]. Previously, it was shown that compound 1 inhibited several viruses belonging to the EV-B species and a clinical isolate of EV-A71, but it failed to inhibit PV-1 and PV-3. Notwithstanding this, the chemical similarity to fluoxetine, as well as the absence of a chiralic center and the CF3 at the phenoxy moiety, make it an interesting candidate for the development of more potent and possibly also broad-spectrum EV inhibitors (Fig 1A). In the present study, we report the rational development of new 2C-targeting antiviral inhibitors based on the backbone of compound 1. We identified several broad-spectrum inhibitors that inhibit representative members of all human EV and RV species tested as well as contemporary isolates of EV-A71, CV-A24v, and EV-D68. Notably, unlike (*S*)-fluoxetine, these broad-spectrum EV inhibitors were shown to be not neuroactive. By raising EV-A71, CV-B3, and EV-D68 variants resistant against one of the broad-spectrum EV inhibitors, we identified some novel resistance mutations in 2C. Employing reverse genetics, we provide evidence for a conserved mechanism of resistance development, involving mutations in the α2 helix of 2C. As these mutations reduced viral replication, additional mutations at distantly localized residues in 2C were acquired to compensate for the loss of fitness. A structural model for a conserved binding pocket in 2C is proposed.

## Results

### Comparison of antiviral activity of (*S*)-fluoxetine and compound 1

We first compared the antiviral spectrum of (*S*)-fluoxetine and compound 1. EVs cause an observable cytopathic effect (CPE), apparent as rounding, detachment, and eventually dying of

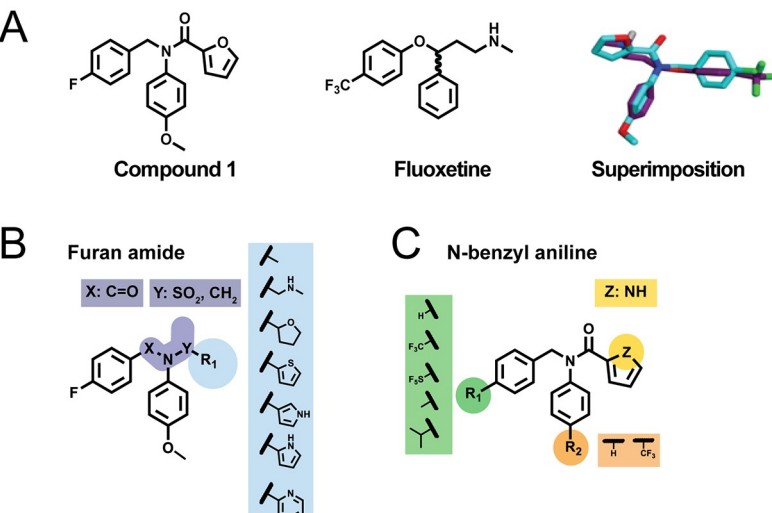

**Fig 1. Structural comparison of fluoxetine and compound 1.** (A) The compounds fluoxetine and compound 1, and a superimposition of them, are depicted. (B) The furan moiety (R1) is highlighted in blue and the amide moiety (X-N-Y) in purple. The substitutions that were explored are shown in the purple and blue boxes. (C) The R1 (green) and R2 (orange) moieties of the N-benzylaniline were substituted to the modifications shown in green and orange boxes. Additionally, the heteroatom (yellow) in the furan moiety was changed.

cells. Both compounds were tested side by side in a CPE-based multicycle viral replication assay to elucidate whether the compounds inhibited replication of several EV serotypes. Subconfluent HeLa R19 cells were treated with 2-fold compound dilutions in the range of 30 μM–0.23 μM. The cells were infected with virus at low multiplicity of infection (MOI) to reach full CPE within 3 days. As previously reported, (S)-fluoxetine inhibited EV-B (CV-B3), EV-D (EV-D68), RV-A (HRV-A2), and RV-B (HRV-B14), but not EV-A (EV-A71) and EV-C (PV-1 and CV-A24) species members [7, 21, 22]. As reported, compound 1 inhibited CV-B3 but not EV-A71 or PV-1 [20]. Additionally, we show antiviral activity of compound 1 against EV-D68 and HRV-B14 but not CV-A24 and HRV-A2. (S)-fluoxetine showed a 50% cytotoxic concentration ($CC_{50}$) of 21.63 ± 1.40, whereas compound 1 did not show any adverse cytotoxic effect in the concentration range tested (Table 1).

## Synthesis of compound 1 analogues

Different structural modifications of compound 1 were planned in order to investigate whether its antiviral activity could be enhanced and its antiviral spectrum broadened. An initial effort was focused on replacing the original furan with different heteroaromatic rings and diverse heterocyclic/aliphatic groups (Fig 1B). The role of the amide bond was also explored by replacing it with either a sulfonamide bond or a methylene bridge. Different substituents on the 4-position on both rings of the N-benzylaniline moiety were also investigated (Fig 1C). Preparation of compound 1 analogues, in which the furan ring and the amide bond were modified, was performed through an efficient two-step synthetic pathway in which compound 4, prepared by reductive amination between 4-fluotobenzaldehyde 2 and p-anisidine 3, was used as common synthetic intermediate. The synthesis routes are displayed in S1 and S2 Figs, and the synthesis route is described in S1 Text. Derivatives 5a–d were synthesized reacting 4 with different acyl or sulfonyl chloride in dichloromethane using trimethylamine as base. Compound 1 and derivatives 5e and 5f—presenting a pyridine and a tetrahydrofuran ring in place of furan, respectively—were obtained through an amide coupling reaction between 4 and the corresponding carboxylic acid in dimethylformamide, using diisopropylethylamine (DIPEA) as base and 2-(1H-benzotriazole-1-yl)-1,1,3,3-tetramethyluronium tetrafluoroborate (TBTU) as a coupling agent. Reductive amination between 4 and furan-2-carbaldehyde yielded compound 5g, whereas preparation of compound 6 was achieved by 2 steps: amide bond formation reacting 4 and 2-bromoacetyl chloride, followed by nucleophilic displacement of the bromine

**Table 1. Comparison of antiviral activity of (S)-fluoxetine and compound 1.**

| Virus | Species | Strain | SFX | Compound 1 |
|---|---|---|---|---|
| *EV-A71* | EV-A | BrCr | >30 | >30 |
| *CV-B3* | EV-B | Nancy | 0.50 ± 0.09 | 1.71 ± 0.07 |
| *PV-1* | EV-C | Sabin1 | >30 | >30 |
| *CV-A24* | EV-C | Joseph | >30 | >30 |
| *EV-D68* | EV-D | Fermon | 0.62 ± 0.22 | 0.32 ± 0.06 |
| *HRV-A2* | RV-A | | 8.92 ± 0.23 | >30 |
| *HRV-B14* | RV-B | | 6.21 ± 0.43 | 10.84 ± 1.27 |
| *$CC_{50}$* | | | 21.56 ± 0,21 | >30 |

Multicycle viral replication assays were performed in HeLa R19 cells, and shown are $EC_{50}$ and $CC_{50}$ values in micromolar. Data represent mean ± SD calculated from at least three different experiments performed in biological triplicates. All underlying experimental data are displayed in S1 Data.

**Abbreviations**: $CC_{50}$, 50% cytotoxic concentration; CV, coxsackievirus; $EC_{50}$, 50% effective concentration; EV, enterovirus; HRV, human rhinovirus; PV, poliovirus; RV, rhinovirus; SFX, (S)-fluoxetine

atom by methyl amine. Different attempts were made for the preparation of compounds 12a–b, in which the *N*-(4-fluoro) benzylaniline portion is bound to position 3 of a pyrrole ring. A coupling reaction either using TBTU or 1,1'-Carbonyldiimidazole (CDI) as coupling agent did not give the desired product, with formation of several undesired species. After failing in converting the pyrrole-3-carboxylic acid to the corresponding acyl chloride using thionyl chloride in dichloromethane, a different approach was applied as reported in S1 Fig synthesis route B. The pyrrole-3-carboxylic acid nitrogen was selectively *tert*-butoxycarbonyl (Boc)-protected through a 3-step synthesis, and the resulting compounds 10a–b were then converted into 11a–b via TBTU-assisted coupling reaction. Removal of the Boc protecting group using trifluoroacetic acid (TFA) in dichloromethane gave compounds 12a–b in a very high yield. Reductive amination between 4-methoxyaniline 3 and furan-2-carbaldehyde 13, followed by the reaction with 4-fluorobenzoyl chloride, gave the final product 15 in a quantitative yield. Analogues 19a–i, bearing different substituents in 4-position of the *N*-benzylaniline moiety, were prepared following the same synthetic pathway adopted for derivatives 5a–d. Reductive amination between differently substituted benzaldehydes and anilines yielded the intermediates 18a–i in a high yield, which were then converted into the corresponding final compounds 19a–i by reaction with furan-2-carbonyl chloride in dichloromethane and trimethylamine as base. 1*H*-pyrrole-2-carbonyl chloride 21, required for the preparation of compound 22, was prepared in situ by refluxing 1*H*-pyrrole-2-carboxylic acid 20 with thionyl chloride, as reported in S2 Fig.

## Antiviral activity of the newly synthesized compounds against CV-B3

We first evaluated the antiviral activity of the synthesized compounds in a multicycle viral replication assay using CV-B3. Substituting the furan amide with a methyl group (5a), a tetrahydrofuran group (5f), or a methyl amine group (6) resulted in a loss of antiviral activity suggesting that the aromatic furan ring is essential for the antiviral activity (S1 Table). Substituting the amide in position Y into a $SO_2$ (5b) or $CH_2$ (5g) also resulted in loss of antiviral activity, suggesting that the amide in position Y is also essential. Changing the methyl group in position X into a carbonyl group and substituting Y into a $CH_2$ (15) also abolished antiviral acitivty. To explore the possibility of changing the furan ring, we substituted the R1 position with several aromatic heterocyclic moieties like thiophen (5d), pyridin (5e), or pyrrole (5c, 12a) (Fig 1B). The thiophene-2-carboxamide (5d: 50% effective concentration [$EC_{50}$] = 0.51 ± 0.05 μM, $CC_{50} \geq 30$ μM) and especially the pyrrole-3-carboxamide substitution (12: $EC_{50}$ = 0.08 ± 0.08 μM, $CC_{50} > 30$ μM) increased the antiviral activity against CV-B3 (approximately 4- and 25-fold, respectively) (S1 Table). Taken together, these data indicated that the amide bond and the furan ring are essential for the antiviral activity, but the modification on the furan moiety can vary.

Next, we explored the substituents R1 and R2 of the N-benzylaniline moiety (Fig 1C). Removal of the R1 substituent (19c), the R2 substituent (19b), or both (19a) had only small effects (i.e., less than 3-fold increase or decrease) on the antiviral activity (S2 Table). Similarly, substitution of R1 to a tri-fluoro moiety (19d, 19e), a methyl group (19g), or a bulky and highly lipophilic electron withdrawing pentafluorosulfanyl (19f), with or without removing the R2 substituent, only marginally affected antiviral activity. Substitution of R2 to a tri-fluoro moiety (19i) also had little effect. Together, this suggests that the substituents in R1 and R2 are not essential for the antiviral activity and can be modified. Only introduction of a branched alkyl group in R1 and removing the R2 substituent (19h) decreased the antiviral activity to a larger extent. Substitution of the R1 group with a CF3 group, removal of the R2 substituent, and substituting the heteroatom in the furan ring to an N (22) also had little, if any, effect on antiviral activity. Taken together, exploring R1 and R2 of the N-benzylaniline moiety results in

active compounds in which the antiviral activity is not drastically increased or decreased compared to compound 1.

## Antiviral effect and cytotoxicity of active analogues in different cell lines

Next, we tested the active compounds for their antiviral effect and cytotoxicity in different cell lines. A CPE-based multicycle viral replication assay using CV-B3 was performed in HeLa R19, HAP1, and Buffalo Green Monkey (BGM) cells. The highest concentration of compounds used was 100 μM. The furan analogues 5c, 12a, and 5e showed no cytotoxicity. Of these compounds, 12a had the most potent antiviral activity, independent of the cell line (S3 Table). The N-benzylaniline analogues 19a, 19b, 19c, 19g, and 22 also did not show any cytotoxicity or very minimal cytotoxicity (19d). The other compounds (5d, 19e, 19f, 19h, 19i) showed cytotoxicity in a similar range as the parental compound 1. Remarkably, 19h showed antiviral activity in the human cell lines HeLa R19 and HAP1 but not in monkey BGM cells. The reason for this is unknown. One explanation is that the drug must be metabolized and that the corresponding enzyme is absent in BGM. Alternatively, the compound may be inactivated or degraded in BGM cells. Overall, compounds 12a, 19b, and 19d showed less cytotoxicity and the strongest increase in selectivity index, which determines the window between antiviral activity and host cell toxicity, in all cell lines tested relative to compound 1.

## Antiviral effect of active compounds against other EVs

To evaluate their spectrum of antiviral activity, we screened the active compounds against a panel of different EVs representative of each of the 4 human EV and of 2 RV species. Many compounds showed a similar antiviral spectrum as the parental compound 1. For some compounds, an extended spectrum of activity was observed. Compounds 19a and 19g showed anti-EV activity but not anti-RV activity. Importantly, compounds 12a, 19b, and 19d inhibited all representative EVs and RVs (Table 2). We also tested the spectrum of activity of 12a and

**Table 2. Antiviral activity of active compounds against representative EV species.**

| Compound | EV-A71 | PV-1 | CV-A24 | EV-D68 | HRV-A2 | HRV-B14 |
|---|---|---|---|---|---|---|
| *1* | >30 | >30 | >30 | 1.13 ± 0.82 | >30 | 12.36 ± 1.51 |
| *5c* | >30 | >30 | >30 | 3.47 ± 0.08 | >30 | >30 |
| *5d* | >30 | >30 | >30 | 2.82 ± 0.40 | >30 | 9.29 ± 0.36 |
| *5e* | >30 | >30 | >30 | 10.45 ± 1.37 | >30 | >30 |
| *12a* | 4.03 ± 0.73 | 3.71 ± 0.18 | 3.51 ± 0.22 | 1.08 ± 0.20 | 8.51 ± 0.22 | 1.45 ± 0.21 |
| *19a* | 3.50 ± 0.31 | 32.58 ± 1.08 | 21.48 ± 0.03 | 1.34 ± 0.45 | >30 | >30 |
| *19b* | 0.89 ± 0.09 | 9.08 ± 0.54 | 6.39 ± 0.74 | 0.50 ± 0.26 | 22.21 ± 0.61 | 10.51 ± 0.61 |
| *19c* | >30 | >30 | >30 | 8.94 ± 0.82 | >30 | >30 |
| *19d* | 0.27± 0.12 | 21.54 ± 0.54 | 19.53 ± 0.55 | 1.37 ± 0.02 | 15.15 ± 1.13 | 4.19 ± 0.58 |
| *19e* | 4.08 ± 0.99 | >30 | >30 | 4.47 ± 0.26 | >30 | 8.38 ± 0.48 |
| *19f* | 4.37 ± 0.50 | >30 | >30 | 14.50 ± 0.86 | >30 | 5.22 ± 0.54 |
| *19g* | 0.60 ± 0.14 | 17.50 ± 1.00 | 10.16 ± 0.87 | 1.49 ± 0.08 | >30 | 12.51 ± 1.05 |
| *19h* | 1.89 ± 0.21 | >30 | >30 | >30 | >30 | 4.64 ± 0.92 |
| *19i* | >30 | >30 | >30 | 3.35 ± 0.19 | >30 | >30 |
| *22* | 1.27 ± 0.49 | >30 | >30 | 12.40 ± 0.81 | >30 | >30 |

Multicycle virus replication assays were performed in HeLa R19 cells, and shown are $EC_{50}$ values in micromolar. Data represent mean ± SD calculated from 2 different experiments performed in biological triplicates. All underlying experimental data are displayed in S1 Data.

**Abbreviations**: CV, coxsackievirus; $EC_{50}$, 50% effective concentration; EV, enterovirus; HRV, human rhinovirus, PV-1, poliovirus 1

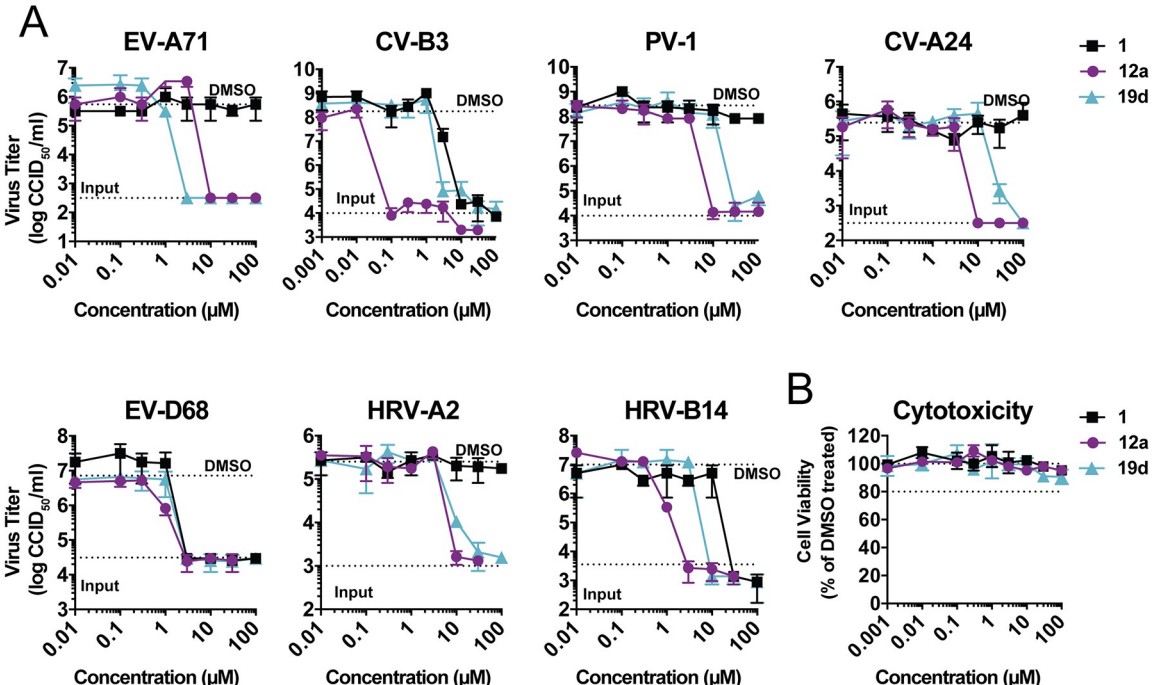

**Fig 2. Antiviral efficacy of compound 1, 12a, and 19d against a panel of EVs.** (A) In a single-cycle viral replication assay, HeLa R19 cells were infected with EV-A71 (strain BrCr), CV-B3 (strain Nancy), PV-1 (strain Sabin), CV-A24 (strain Joseph), EV-D68 (strain Fermon), HRV-A2, and HRV-B14 at MOI 1. At 30 minutes after infection, the cells were treated with serial dilutions of the parental compound 1 and the analogues 12a and 19d. Eight hours (EV-A71, CV-B3, PV-1, CV-A24) or ten hours (EV-D68, HRV-2, HRV-14) post infection, cells were freeze-thawed three times, and virus titers of lysates were determined by endpoint dilution. (B) In parallel, uninfected cells were treated with compounds only, and cell viability was determined using an MTS assay. Data represent mean ± SD from one representative experiment. Every experiment was performed in biological triplicates, and two independent experiments were performed. All underlying experimental data are displayed in S2 Data. CCID$_{50}$, cell culture infectious dose; CV, coxsackievirus; EV-A71, enterovirus A71; EV-D68, enterovirus D68; HRV, human rhinovirus; MOI, multiplicity of infection; MTS, 3-(4,5-dimethylthiazol-2-yl)-5-(3-carboxymethoxyphenyl)-2-(4-sulfophenyl)-2H-tetrazolium; PV-1, poliovirus 1.

19d in a single-cycle viral replication assay. Consistent with the results in the multicycle viral replication assay, compounds 12a and 19d inhibited all representative viruses in a single-cycle viral replication assay (Fig 2A), and none of them showed any cytotoxic effect (Fig 2B). Notably, 12a showed higher potency against all viruses tested except against EV-A71.

The broad-spectrum EV and RV inhibitors 12a, 19b, and 19d were further profiled for their broad-range antiviral activity against several contemporary isolates of EV-A71 (isolates derived from an outbreak of hand-foot-and-mouth disease in Taiwan), CV-A24v (isolates derived from an outbreak in Malaysia 2002–2003), and EV-D68 (isolated from patients with respiratory infection in the Netherlands 2009–1010) [4, 28, 29]. As expected, the parental compound 1 only inhibited the contemporary strains of EV-D68 (Table 3). (S)-fluoxetine also inhibited contemporary strains of EV-D68 and some, but not all, contemporary EV-A71 strains. Importantly, 12a, 19b, and 19d inhibited all clinical isolates of EV-A71, CV-A24v, and EV-D68 tested (Table 3). Taken together, we identified three compounds—12a, 19b, and 19d—that inhibited all EVs, all RVs, and all clinical isolates of the serotypes EV-A71, CV-A24, and EV-D68.

## Sensitivity of CV-B3 2C mutants to the broad-spectrum EV and RV compounds

Previously, it was shown that compound 1–resistant CV-B3 viruses had acquired 2C mutations S58N, C179F, I227V, and N257D. Subsequently, it was shown that C179F alone could provide

**Table 3. Antiviral activity of active compounds against different clinical isolates of EV-A71, EV-D68, and CV-A24v.**

| Virus | Strain | Cluster | SFX | 1 | 12a | 19b | 19d |
|---|---|---|---|---|---|---|---|
| *EV-A71* | BrCr | A | >30 | >30 | 4.48 ± 0.42 | 1.12 ± 0.15 | 1.68 ± 0.62 |
| | TW/70811/08 | B5 | >30 | >30 | 8.18 ± 0.61 | 0.92 ± 0.17 | 0.754 ± 0.01 |
| | TW/96016/08 | B5 | >30 | >30 | 9.47 ± 0.04 | 1.89 ± 0.02 | 1.94 ± 0.82 |
| | TW/72232/04 | C4 | 6.20 ± 1.45 | >30 | 6.41 ± 2.09 | 3.83 ± 0.25 | 2.54 ± 1.72 |
| | TW/2728/04 | C4 | 4.93 ± 0.57 | >30 | 8.33 ± 0.65 | 2.10 ± 0.08 | 2.33 ± 1.23 |
| | TW/2945/98[a] | NA | 4.65 ± 1.11 | >30 | 9.17 ± 0.41 | 2.02 ± 0.15 | 2.9 ± 1.03 |
| *EV-D68* | Fermon | | 0.74 ± 0.12 | 0.65 ± 0.20 | 1.23 ± 0.16 | 0.58 ± 0.10 | 1.36 ± 0.22 |
| | 4311000742 | A | 0.66 ± 0.14 | 3.83 ± 0.40 | 1.86 ± 0.16 | 1.31 ± 0.90 | 3.91 ± 0.13 |
| | 4310901348 | A | 1.24 ± 0.32 | 4.47 ± 0.13 | 2.72 ± 0.96 | 1.71 ± 0.68 | 3.55 ± 0.43 |
| | 4310902042 | B | 0.57 ± 0.30 | 4.05 ± 0.65 | 1.44 ± 0.23 | 0.91 ± 0.92 | 3.53 ± 0.41 |
| | 4310900947 | B | 0.95 ± 0.27 | 3.65 ± 0.34 | 1.84 ± 0.40 | 1.85 ± 0.18 | 4.60 ± 0.37 |
| | 4310902284 | C | 1.46 ± 0.36 | 2.23 ± 0.12 | 2.31 ± 0.82 | 1.29 ± 0.21 | 1.82 ± 0.57 |
| *CV-A24* | Joseph | | >30 | >30 | 3.45 ± 0.42 | 4.99 ± 0.61 | 18.65 ± 0.67 |
| *CV-A24v* | 110386 | | >30 | >30 | 1.94 ± 0.07 | 2.53 ± 0.45 | 15.66 ± 1.32 |
| | 110389 | | >30 | >30 | 3.77 ± 0.32 | 5.65 ± 0.23 | 18.55 ± 0.44 |
| | 110390 | | >30 | >30 | 1.91 ± 0.55 | 4.16 ± 0.17 | 16.75 ± 0.33 |
| | 110392 | | >30 | >30 | 2.32 ± 0.23 | 4.35 ± 0.18 | 19.06 ± 0.97 |

[a]Provided by Steve Oberste from the Centers of Disease Control and Prevention [20].

Multicycle viral replication assay was performed in HeLa R19 cells, and shown are $EC_{50}$ values in micromolar. Data represent mean ± SD calculated from 2 different experiments performed in biological triplicates. All underlying experimental data are displayed in S1 Data.

**Abbreviations**: CV, coxsackievirus; $EC_{50}$, 50% effective concentration; EV, enterovirus; NA, not available; SFX, (*S*)-fluoxetine

protection against compound 1 (other mutants were not tested) [20]. Moreover, mutations C179F, I227V, and the triple-mutant A224V-I227V-A229V, as well as F190L, conferred resistance to (*S*)-fluoxetine [20, 24]. This likely suggests that both compounds target a common binding pocket. We investigated whether these mutations conferred cross-resistance to 12a and 19b. For this, we used a recombinant CV-B3 virus encoding a *Renilla* luciferase reporter gene (Rluc-CV-B3) upstream of the capsid coding region. Cells were infected with Rluc-CV-B3 viruses harboring 2C mutations and treated with serial dilutions of compound. Eight hours post infection, *Renilla* luciferase was determined as a sensitive and quantitative read-out for virus replication. As previously reported, the mutation C179F conferred resistance to compound 1 [20]. Additionally, we observed that all other common 2C resistance mutations (C179Y, F190L, I227V, and A224V-I227V-A229V) conferred compound 1 resistance (Fig 3A). The resistance profile of 12a resembled that of the parental compound 1 and (*S*)-fluoxetine (Fig 3B). Remarkably, I227V and A224V-I227V-A229V did not confer resistance to 19d, whereas C179F and F190L provided only a low level of resistance.

## Raising resistant viruses against compound 19d

As known 2C mutations only marginally provided resistance to 19d, we raised 19d-resistant virus variants of EV-A71, CV-B3, and EV-D68 via a clonal resistance selection procedure described previously [30].

**CV-B3.** Two 19d-resistant CV-B3 virus pools were obtained. Both contained a mutation at position F190 (F190L or F190V) together with other mutations (Table 4). The first CV-B3 virus pool acquired the mutation F190V and three additional 2C mutations: A220P, S233P, and D234N (Table 4 and Fig 4A). A220P is located close to the motif C that is part of the

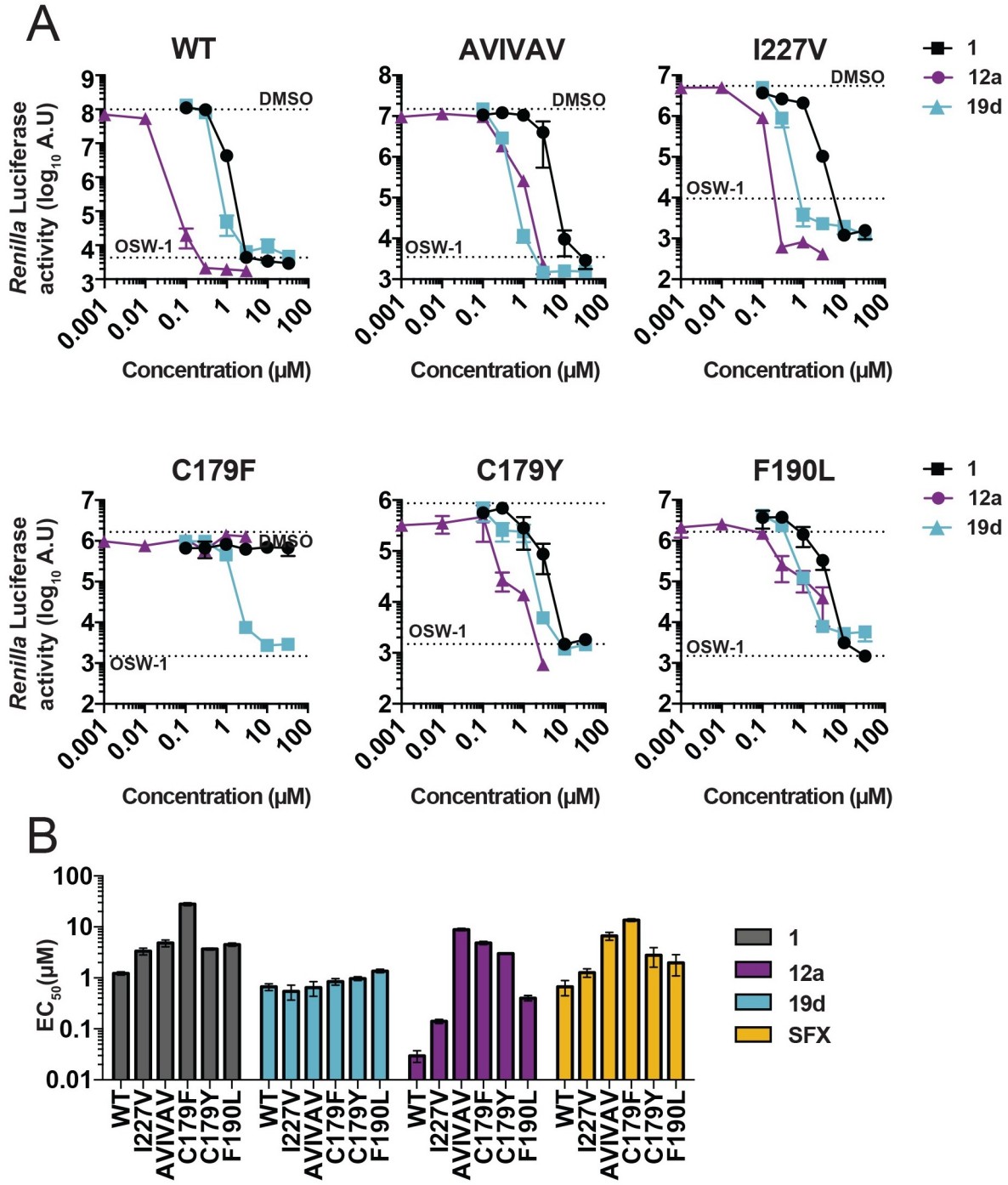

**Fig 3. Mutations in the CV-B3 protein 2C confer resistance to compound 1 and 12a but not 19d.** In a single-cycle viral replication assay, HeLa R19 cells were infected with a selection of Rluc-CV-B3 reporter viruses containing previously identified mutations in the nonstructural protein 2C conferring resistance to (S)-fluoxetine [24]. (A) HeLa R19 were infected with an MOI 0.1 of Rluc-CV-B3 WT virus, the triple mutant (A224V-I227V-A229V, designated AVIVAV), the I227V single mutant, the C179F or the C179Y mutant, and the F190L mutant. At 30 minutes post infection, the cells were treated with serial dilutions of compound 1, 12a, 19d, as well as 10 nM OSW-1 as a control replication inhibitor that acts via the host protein OSBP. Eight hours post infection, cells were lysed, and virus titers were determined by measuring Rluc activity as a quantitative measurement for viral replication. Data represent mean ± SD from one experiment representative of three independent experiments performed in biological triplicates. (B) The EC50 values of the three independent experiments were calculated for each mutant Rluc-CV-B3 virus and each compound. (S)-fluoxetine was used as a positive control. All underlying experimental data that are displayed can be found in S2 Data. A.U., arbitrary units; AVIVAV, CV-B3 virus harboring the triple

mutation A224V-I227V-A229V; CV, coxsackievirus; EC$_{50}$, 50% effective concentration; MOI, multiplicity of infection; OSBP, oxysterol-binding protein; Rluc, *Renilla* luciferase; WT, wild type.

ATPase domain and structurally immediately followed by the 224AGSINA229 loop, a hotspot for resistance mutations toward 2C inhibitors [31, 32]. The second CV-B3 virus pool contained the mutations F190L and H243R (Table 4). The latter amino acid is located near the arginine finger (which is formed by 240R and R241) of 2C. Both resistant virus pools are cross-resistant to 12a, 19b, and (S)-fluoxetine. We reverse engineered the single mutations observed in pool 2 (F190L and H243R), either alone or in combination, into the Rluc-CV-B3 reporter virus. Viruses were characterized for their 19d sensitivity in a single-cycle viral replication assay, and replication kinetics were analyzed using *Renilla* luciferase as read-out. Mutation F190L conferred a low level of resistance to 19d, whereas H243R did not provide any resistance. Remarkably, introducing both mutations F190L/H243R conferred a high level of resistance (Fig 4B). Replication analysis showed that each of the single mutations caused a delay in replication kinetics but that the double mutation restored replication kinetics to wild-type levels (Fig 4C). Mapping of the mutations on the CV-B3 2C homology model that we previously published (24) showed that residue F190 is located in the α2 helix, whereas residue H243 is localized more distantly (Fig 4D and S3 Fig). Together, these data suggest that CV-B3 first acquired a mutation at position F190, which gives a low level of resistance and maps in the conserved part of the α2 helix of the CV-B3 homology model [33]. However, mutation of this position resulted in a loss of viral fitness. Other mutations in more distantly located residues (e.g., H243R) were then acquired that further increased resistance and restored the replication fitness of the virus.

**EV-A71.** Four different 19d-resistant EV-A71 virus pools were obtained. All contained a mutation at position M193, either M193L or M193F, alone or in combination with additional mutations (Table 4 and Fig 4A). M193L was previously reported to provide resistance to guanidine hydrochloride (GuaHCl), a well-known 2C inhibitor [34]. Two virus pools contained a second mutation, R151H, and displayed a further increase in resistance. The fourth virus pool contained M193L, R151H, and a third mutation in the Walker A motif, T133S. This pool exhibited only a slight increase in 19d resistance. Similar profiles were observed when these

**Table 4. Phenotypic characterization of 19d-resistant virus pools.**

| Virus | Genotype 2C | 19d | Fold | 12a | Fold | 19b | Fold | SFX | Fold |
|---|---|---|---|---|---|---|---|---|---|
| *EV-A71* | WT | 0.66 ± 0.09 | 1 | 4.65 ± 0.27 | 1 | 0.82 ± 0.12 | 1 | >30 | ND |
| *EV-A71 pool 1* | M193L | 3.14 ± 0.36 | 5 | 3.34 ± 0.20 | 1 | 2.27 ± 0.1 | 3 | >30 | ND |
| *EV-A71 pool 2* | R151H, M193L | 7.64 ± 0.27 | 12 | 20.48 ± 1.74 | 4 | 10.74 ± 0.40 | 13 | >30 | ND |
| *EV-A71 pool 3* | R151H, M193F | 6.97 ± 0.07 | 11 | 23.85 ± 1.34 | 5 | 11.10 ± 0.46 | 14 | >30 | ND |
| *EV-A71 pool 4* | T133S, R151H, M193L | 8.24 ± 0.16 | 12 | 25.34 ± 0.90 | 5 | 11.59 ± 2.01 | 14 | >30 | ND |
| *CV-B3* | WT | 1.26 ± 0.15 | 1 | 0.02 ± 0.01 | 1 | 0.29 ± 0.05 | 1 | 0.43 ± 0.16 | 1 |
| *CV-B3 pool 1* | F190V, A220P, S233P, D234N | 7.44 ± 0.49 | 5 | 0.34 ± 0.12 | 18 | 1.46 ± 0.31 | 5 | 4.58 ± 0.31 | 11 |
| *CV-B3 pool 2* | F190L, H243R | 12.40 ± 1.13 | 9 | 0.26 ± 0.03 | 14 | 2.18 ± 0.03 | 8 | 4.13 ± 1.94 | 10 |
| *EV-D68* | WT | 1.55 ± 0.17 | 1 | 1.25 ± 0.09 | 1 | 0.83 ± 0.09 | 1 | 0.65 ± 0.16 | 1 |
| *EV-D68 pool 1* | T196S | 3.79 ± 0.18 | 2 | 3.12 ± 0.32 | 2 | 1.75 ± 0.10 | 2 | 10.21 ± 0.01 | 16 |
| *EV-D68 pool 2* | V96M | 4.35 ± 0.09 | 3 | 4.18 ± 0.03 | 3 | 2.76 ± 0.14 | 4 | 10.05 ± 0.45 | 15 |

Multicycle viral replication assays were performed in HeLa R19 cells, and shown are EC$_{50}$ values in micromolar and fold resistance. Data represent mean ± SD calculated from three different experiments all performed in biological triplicates. All underlying experimental data are displayed in S1 Data.

**Abbreviations**: CV, coxsackievirus; EC$_{50}$, 50% effective concentration; EV, enterovirus; ND, not determined; SFX, (S)-fluoxetine; WT, wild type

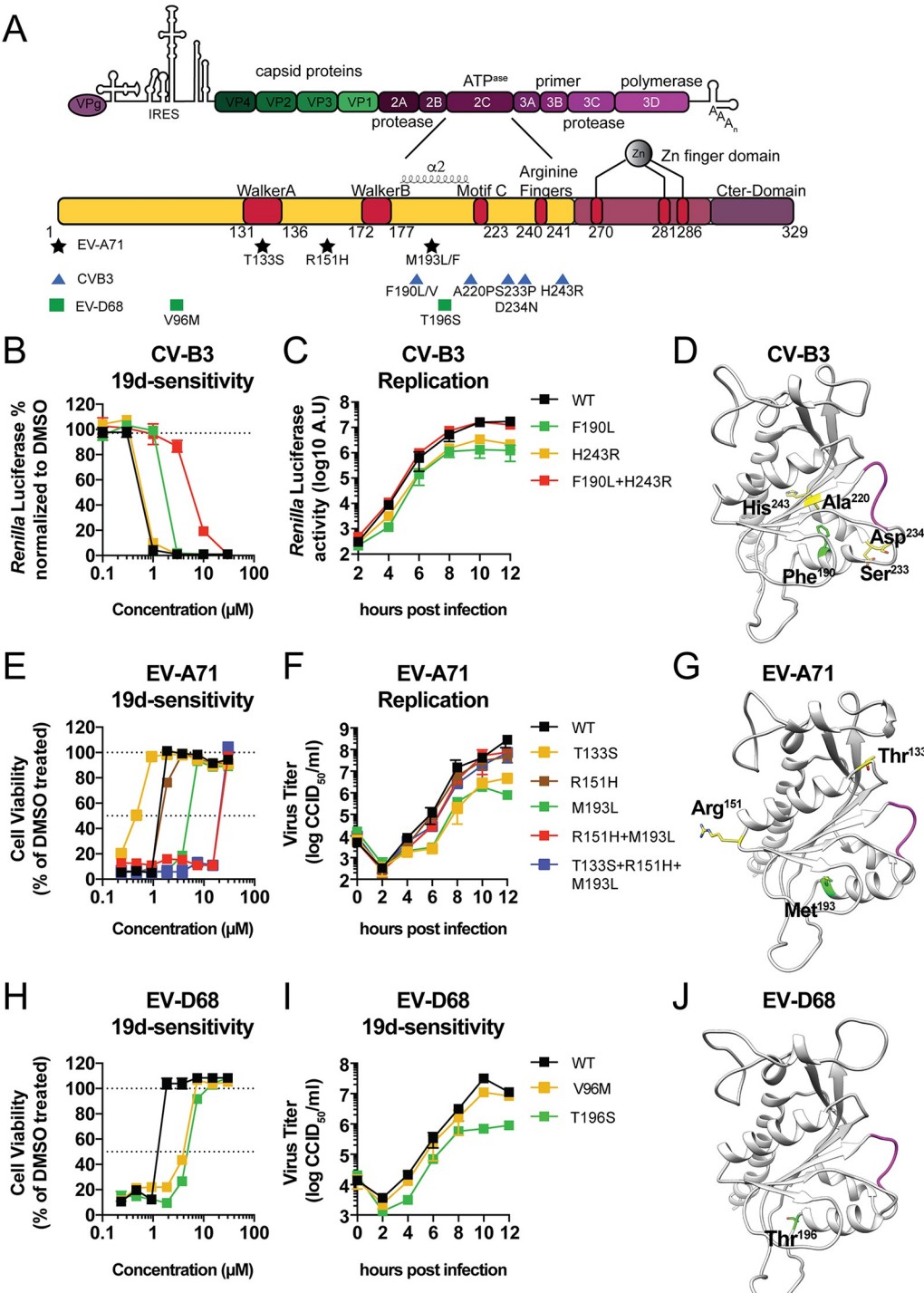

**Fig 4. Effect of reverse-engineered 2C mutations on 19d sensitivity and replication fitness.** (A) Schematic representation of the viral genome with a focus on the 2C protein and the functional domains. The resistance mutations for EV-A71 are highlighted with black stars, CV-B3 resistance mutations are depicted in blue triangles, and EV-D68 resistance mutations in green squares. (B) Single-cycle viral replication assay to determine 19d sensitivity of Rluc-CV-B3 reporter viruses harboring 2C mutations. HeLa R19 cells were infected with MOI 0.1 of RLuc-CV-B3 reporter virus and treated with serial dilutions of 19d. Rluc activity was determined at 8 hours post infection as a quantitative measure of replication. (C) Replication of Rluc-CV-B3 2C mutant viruses. HeLa R19 cells were infected at MOI 0.1 with the RLuc-CV-B3 viruses, and at the indicated time points, the cells were lysed and luciferase activity was determined. (D) The 2C mutations of CV-B3 2C which were acquired during resistance selection are mapped onto the previously published

homology model of CV-B3 2C [24]. The amino acid F190 is highlighted in green, the amino acids H243 and A220P in yellow, and the amino acids S233 and D234 in red. (E) Multicycle viral replication assay to determine 19d sensitivity of reverse-engineered EV-A71 2C mutant viruses. HeLa R19 cells were treated with serial dilutions of 19d and infected with MOI of 0.001 of the reverse-engineered EV-A71 mutant viruses. After 3 days, the cells' viability was determined using an MTS assay. (F) Replication of EV-A71 reverse-engineered viruses. After infection for 30 minutes at MOI 5, cells were incubated for the indicated time points. Cells were freeze-thawed three times to harvest infectious virus particles. Total virus titers were determined by endpoint dilution. (G) The 2C mutations of EV-A71 2C which were acquired during resistance selection are mapped onto the EV-A71 2C crystal structure PDB: 5GRB [33]. The amino acid M193 is depicted in green, the amino acid R151 in yellow, and T133 in red. (H) Multicycle viral replication assay to determine 19d resistance of reverse-engineered EV-D68 2C mutant viruses. The experiment was done similar to (E), and cells were infected with MOI 0.1 to reach full CPE within 3 days. (I) Growth curves of reverse-engineered EV-D68 viruses. Growth kinetics was assessed similar to in (C). (J) The amino acid T196 of EV-D68 2C is highlighted in green and mapped onto a homology model of EV-D68 2C which was built similar to the CV-B3 homology model based on the crystal structure of EV-A71 (PDB: 5GRB). The primary mutation T196S is highlighted in green. Data represent mean ± SD from one experiment representative of at least two independent experiments performed in biological triplicates. All underlying experimental data that are displayed can be found in S2 Data. A.U., arbitrary units; CCID$_{50}$, 50% cell culture infective dose; CV, coxsackievirus; EV, enterovirus; IRES, internal ribosomal entry side; MOI, multiplicity of infection; MTS, 3-(4,5-dimethylthiazol-2-yl)-5-(3-carboxymethoxyphenyl)-2-(4-sulfophenyl)-2H-tetrazolium; PDB, protein database; Rluc, *Renilla* luciferase; WT, wild type.

virus pools were tested for their resistance to inhibitors 12a and 19b (Table 4). To obtain more insight into the effects of the mutations on 19d sensitivity and virus growth, we introduced them—either individually or in combination—into the infectious clone of EV-A71 by reverse genetics. The mutant viruses were characterized for their 19d resistance in a multicycle viral replication assay (Fig 4E). Moreover, we determined single-cycle replication kinetics of these viruses in the absence of inhibitor (Fig 4F). The single mutation M193L, but not T133S or R151H, conferred 19d resistance. Whereas R151H alone did not confer resistance, the double mutation (M193L/R151H) further increased the resistance. Introduction of a third mutation, T133S (M193L/R151H/T133S), slightly further increased resistance (Fig 4E). In the replication kinetics analysis, we observed that replication of viruses carrying single mutations M193L or T133S, but not R151H, was impaired (Fig 4F). Remarkably, viruses carrying double mutation M193L/R151H or triple mutation M193L/R151H/T133S showed wild-type replication kinetics. These resistance mutations were mapped on the EV-A71 2C crystal structure (PDB: 5GRB) (Fig 4G). The data are in line with those observed for CV-B3 in that a primary mutation in the α2 helix of EV-A71 2C (M193L) was acquired that provided resistance to 19d but which was at the expense of virus fitness. Additional mutations (R151H and T133S) were then acquired which compensated for the loss of fitness and further increased resistance.

**EV-D68.** Two 19d-resistant EV-D68 pools were obtained. One pool contained the single mutation V96M, while the other contained the single mutation T196S (Fig 4A). Both resistant virus pools conferred 19d resistance and cross-resistance to 12a, 19b, and (*S*)-fluoxetine (Table 4). Both mutations were reverse engineered into an EV-D68 infectious clone and tested for their 19d sensitivity in a multicycle viral replication assay as well as for their replication kinetics. Both V96M and T196S provided 19d resistance, although resistance was rather moderate (Fig 4H). The V96M mutant virus grew comparably to wild-type, whereas the replication kinetics of T196S was impaired (Fig 4I). Mapping of the mutations on an EV-D68 2C homology model that we built on the 2C crystal structure of EV-A71 showed that residue T196 is located very close to the α2 helix (Fig 4J). Unfortunately, we cannot draw any conclusions on the position and/or role of residue V96 since structural data of the first 116 amino acids of 2C are missing.

In summary, we observed a common pattern that 19d-resistant EV-A71, CV-B3, and EV-D68 variants acquired a mutation in the α2 helix or very close to it. Acquiring these mutations came with a fitness cost for the virus. Additional mutations in more distantly localized

residues were obtained that increased resistance and improved the viral fitness. Together, this suggests that the α2 helix may form a part of the compound binding site.

## In silico identification of a putative solvent accessible tunnel in 2C

To strengthen the hypothesis that the α2 helix is likely part of the binding pocket, we used computational methods to identify solvent accessible tunnels within the EV-A71 2C crystal structure (PDB: 5GRB) [35–38]. Using the amino acid M193 in the α2 helix of EV-A71 2C as starting coordinate, and an origin radius of 5 Å, we identified three solvent accessible tunnels surrounding the α2 helix in EV-A71 (Fig 5A). These tunnels connect the interior of the molecule with the surrounding environment and provide a possible entry side for compounds to bind 2C.

Next, we explored the occurrence of solvent accessible tunnels in the 2C protein of PV, of which also a crystal structure has been solved (PDB: 5Z3Q) [33]. Previous studies have identified 2C mutations F164Y, N179G/A, and M187L in PV to confer resistance to the 2C inhibitors GuaHCl or MRL-1237 [35–39]. These residues localize very close to the α2 helix (Fig 5B). We mapped the mutations on the 2C crystal structure of PV (PDB: 5Z3Q) and looked for solvent accessible tunnels using M187 amino acid as starting coordinate and an origin radius of 5 Å. We were able to identify a tunnel at the α2 helix of 2C PV, and this tunnel shows similarities to the tunnel we observed in EV-A71 2C. The same tunnel was predicted when residues F164 (S4 Fig Panel A) or N179 (S4 Fig Panel B) were used as starting coordinate. Additionally, other programs also predicted very similar solvent-accessible tunnels or cavities within 2C of EV-A71 and PV-1 (S4 Fig Panel C-F). Taken together, we propose that the α2 helix in 2C is part of the binding pocket of inhibitors and that mutations in this helix disrupt binding of these inhibitors, thereby providing a first layer of resistance.

To test the hypothesis that the 2C α2 helix mutation disrupted compound binding, we performed thermal shift assays with (S)-fluoxetine and 19d. We expressed recombinant fragments of CV-B3 2C lacking the first 116 amino acids and the corresponding 2C fragment of the single mutant F190L. Both (S)-fluoxetine and 19d caused a dose-dependent shift in the melting temperature of the wild-type 2C protein, indicative of a direct binding of the wild-type protein to

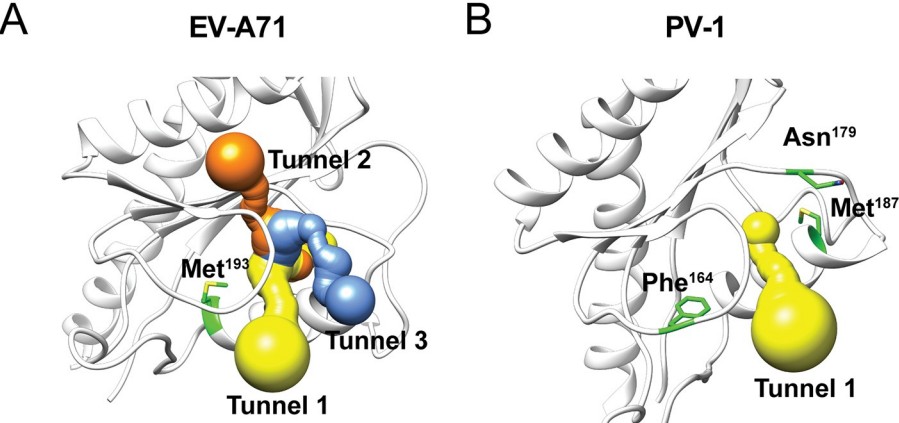

**Fig 5. In silico prediction of solvent accessible tunnels reveals a binding pocket surrounding the α2 helix of 2C.**
(A) The MOLE tool was used to identify solvent exposed tunnels surrounding the α2 helix of 2C [40]. For EV-A71 the 2C crystal structure PDB: 5GRB was used, and the amino acid M193 was used as starting point [33]. (B) For PV the 2C crystal structure PDB: 5Z3Q with the amino acid M187 as starting point [41]. The primary mutations of EV-A71 and PV-1 are highlighted in green and the in silico predicted tunnels are colored in yellow, blue, and orange. Asn, asparagine; EV, enterovirus; PDB, protein database; Phe, phenylalanin; PV, poliovirus.

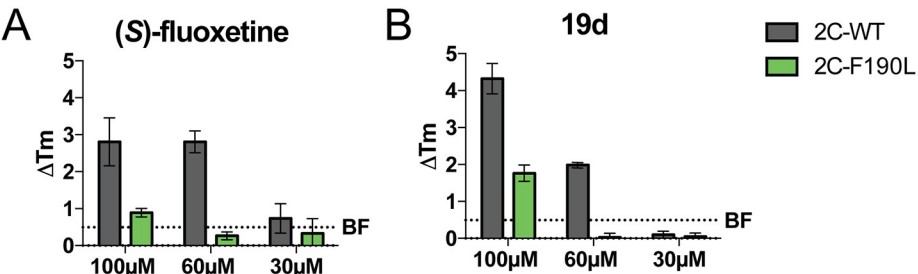

**Fig 6. Resistance mutations in the α2 helix of CV-B3 2C disrupt compound binding.** The binding of (A) (*S*)-fluoxetine and (B) 19d to a recombinant fragment of the CV-B3 2C protein and the CV-B3 2C α2 helix mutant F190L was assessed by thermal shift assays. The thermal shift of 2C is represented by change in melting temperature ($\Delta T_m$). The dashed line represents data from the negative control BF738735, an inhibitor of the pan-EV host factor phosphateidylinositol-4-III β, used at a concentration of 100 μM. Data shown are representative of one out of two experiments which were performed in technical triplicates. Error bars depict SD calculated from technical triplicates of the representative experiment. All underlying experimental data that are displayed can be found in S2 Data. CV, coxsackievirus; EV, enterovirus; Tm, melting temperature; WT, wild type.

the compounds (Fig 6A and 6B). No shift was observed upon testing the F190L mutant, indicating that the binding of the compounds was disrupted by the α2 helix mutation. These data provide the first evidence that mutations in the α2 helix of 2C disrupt compound binding.

## Combination of 12a and 19b reveals most potent broad-spectrum EV and RV inhibitor

Our data suggest that 12a and 19b as well as 19d target the same common druggable binding pocket in the 2C proteins of CV-B3, EV-A71, and EV-D68. Next, we were wondering whether a combination of the chemical features of the broad-spectrum EV and RV inhibitors 12a and 19b could increase the antiviral activity. Therefore, we synthesized a new compound with the combined chemical moieties of the furan amide analogue 12a with the fluoride moiety of the N-benzylaniline analogues 19b (Fig 7A). The antiviral activity and the antiviral spectrum of the new combined analogue 12b was tested in a multicycle viral replication assay against the panel of prototypic EV and RV species. Compound 12b showed an improved antiviral profile against all tested viruses compared to the parental compound 1 and the analogue 12a (Fig 7B). The antiviral activity against the serotypes CV-B3, EV-D68, and HRV-B14 is in the nM range, whereas its activity is in the sub-μM or low μM range against EV-A71, PV-1, CV-A24, and HRV-2. These data show that broad-spectrum 2C inhibitors can be developed in the therapeutically relevant nM range.

## Broadly active EV inhibitors do not affect serotonin, dopamine, or norepinephrine transporter activity and neural activity

Fluoxetine is an SSRI, and both enantiomers are equipotent in SSRI activity. Given the structural similarity of the compound 1 analogues to fluoxetine, we wanted to exclude that these analogues also exhibit the same undesirable neurological effects. To do this, we investigated whether the compounds inhibit the monoamine reuptake of serotonin, dopamine, and norepinephrine through inhibition of their corresponding transporters serotonin transporter (SERT), dopamine transporter (DAT), and norepinephrine transporter (NET). Previously, it was shown using the Neurotransmitter Transporter Uptake Assay that human embryonic kidney (HEK) 293 cells transfected with human DAT, NET, and SERT show a stable and near linear increase in fluorescence indicating proper transporter function (Fig 8A) [42]. Exposure of

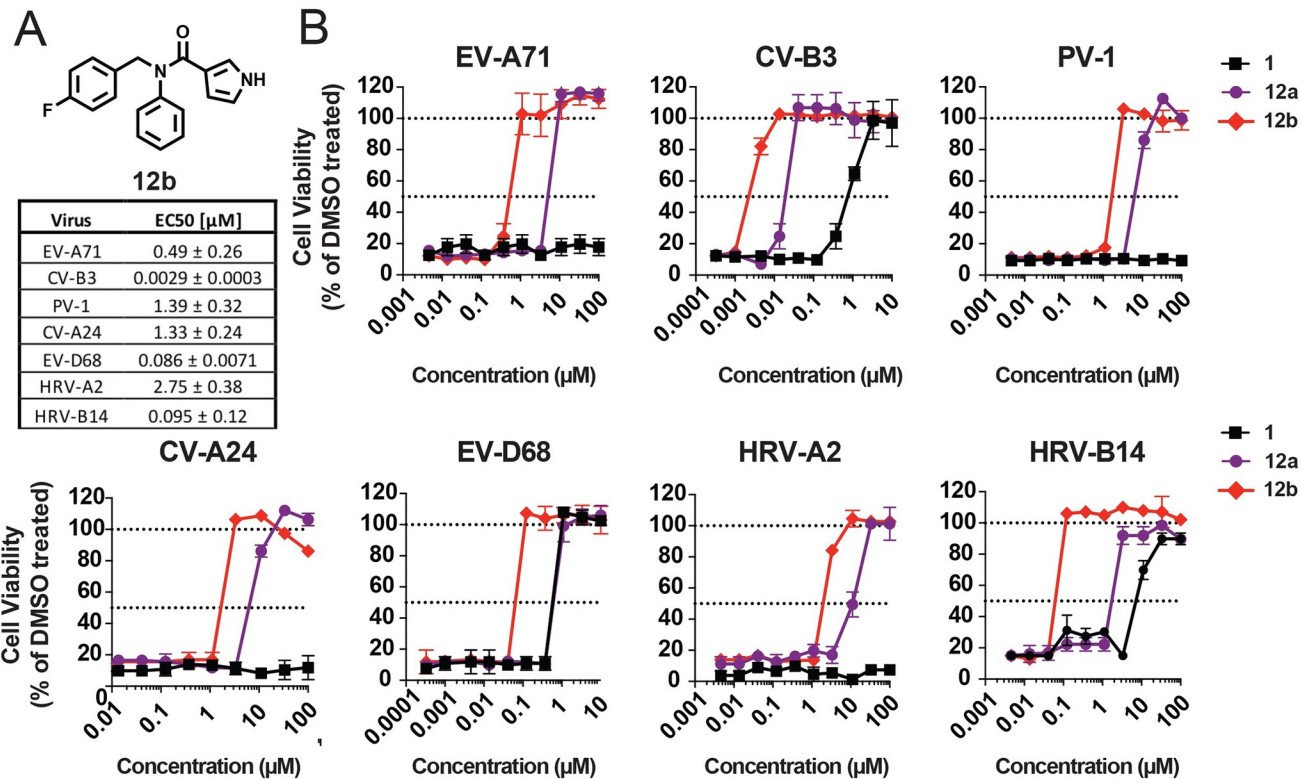

**Fig 7. Combination of compound 12a and 19b reveals the most potent broad-spectrum EV and RV inhibitor 12b.** (A) The chemical backbone of 12b is shown. The table represents the $EC_{50}$ values in μM ± SD of compound 12b. $EC_{50}$ is calculated for the panel of representative EV and RV. $EC_{50}$ values represent mean values that are calculated from three independent experiments which were performed in biological triplicates. (B) A multicycle viral replication assay was performed to determine the antiviral activity of compound 12b. HeLa R19 cells were treated with serial dilutions of parental compound 1, the furan amide analogue 12a and 12b. Immediately thereafter, cells were infected with EV-A71 (BrCr), CV-B3 (Nancy), PV-1, CV-A24 (Joseph), EV-D68 (Fermon), HRV-A2, and HRV-B14 at low MOI (depending on the virus, see Material and methods) to reach full CPE within 3 days. Data are shown from one representative experiment out of three independent experiments which were performed in biological triplicate. All underlying experimental data that are displayed can be found in S2 Data. CPE, cytopathic effect; CV, coxsackievirus; $EC_{50}$, 50% effective concentration; EV, enterovirus; HRV, human rhinovirus; PV, poliovirus; RV, rhinovirus.

HEK293 cells expressing human SERT to fluoxetine (1 and 100 μM) resulted in complete inhibition of SERT-mediated uptake (Fig 8B). These results are in line with fluoxetine acting as an SSRI and with previous results showing concentration-dependent inhibition of human SERT by fluoxetine (50% inhibitory concentration [$IC_{50}$] = 0.3 μM, full inhibition at $\geq$ 1 μM [42]). As reported, fluoxetine promiscuously inhibited also the DAT and NET activity at 100 μM. In contrast, neither 12a, 19b, 19d, nor the combined compound 12b inhibited SERT, DAT, or NET function at 1 μM or 100 μM, respectively. We also used a multiwell micro-electrode array (mwMEA) to assess effects of fluoxetine and the compound 1 analogues on spontaneous neuronal activity to determine their neurotoxic potential (for review see [43] and overview of methods see S5 Fig). Unlike fluoxetine—which inhibited the mean spike rate (MSR), mean burst rate (MBR), and mean network burst rate (MNBR) in a concentration-dependent manner—none of the compound 1 analogues affected neuronal activity (S6 Fig).

## Discussion

EVs, especially EV-A71, EV-D68, and CV-A24v, impose serious public health threats. Currently, there are no antiviral therapies licensed to treat EV infections. The highly conserved nonstructural protein 2C fulfills pleiotropic functions during the viral life cycle. Hence, it

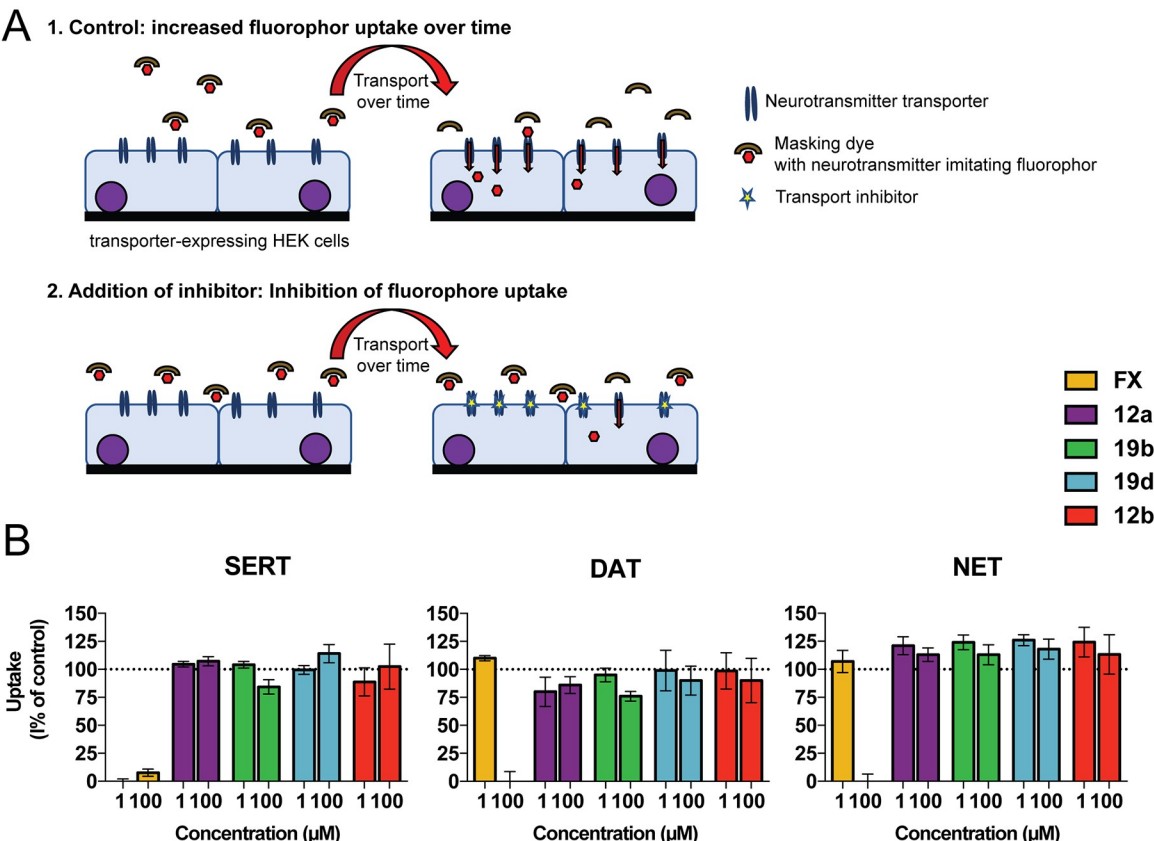

**Fig 8. Broadly active EV inhibitors do not affect SERT, DAT, or NET activity.** (A) Overview of the principle of the fluorescent measurements of neurotransmitter transporter function. Cells expressing human monoamine transporters (SERT, DAT, or NET) are incubated with a fluorescent transporter substrate and a cell-impermeable masking dye that extinguishes only extracellular fluorescence (left). Cells gain fluorescence following uptake of the substrate, and the increase in fluorescence is a measure for transporter function (top right). Cells with impaired transporter function will take up less substrate, resulting in lower fluorescence (bottom right). (B) Effects of fluoxetine and compounds 12a, 12b, 19b, and 19d on inhibition of SERT-, DAT-, and NET-mediated uptake of fluorescent substrate. Uptake is shown as mean ± SEM as percentage compared to control (DMSO) wells ($n$ = 7–16 wells, derived from 2 independent cultures). All underlying experimental data that are displayed can be found in S2 Data. DAT, dopamine transporter; EV, enterovirus; FX, racemic fluoxetine; NET, norepinephrine transporter; SERT, serotonin transporter.

represents an interesting target for the development of broad-spectrum anti-EV inhibitors. The FDA-approved drug fluoxetine, which is an SSRI, was an attractive candidate for the treatment of EV infections, but several recent studies raised concerns about its clinical application. One retrospective study revealed adverse effects when using fluoxetine as treatment option for EV-D68-associated paralysis. The data suggested a worsening of patient condition, possibly due to its SSRI activity [26]. Additionally, our previous work showed that the antiviral activity of fluoxetine is unlikely to be decoupled from its SSRI activity [24, 27]. This clearly demonstrates that more potent and safer molecules are needed for treatment of EV infections.

In a high-throughput screen, compound 1 was identified as a CV-B3 inhibitor [20]. The chemical structure is reminiscent of fluoxetine but lacks a chiral center and the tri-fluoro moiety important for the SSRI activity of fluoxetine. In this study, we investigated the pharmacophoric features of compound 1 which underpin its antiviral activity. An important focus was to explore the furan amide moiety and the substituents of the N-benzylaniline moiety. First, the synthesized analogues were screened for their antiviral activity against CV-B3. The structure-activity relationship study revealed that the furan amide, but not the N-benzylaniline

moiety, was essential for the antiviral activity. Even though the furan amide was essential, different modifications can be introduced. Subsequently, compounds that inhibited CV-B3 were evaluated for their ability to inhibit representatives of four human EV and two RV species. Several compounds showed an increased potency and broadened antiviral spectrum relative to parental compound 1. Four of these compounds—the furan analogues 12a and 12b and the N-benzylaniline analogues 19b and 19d—inhibited all representative EVs and RVs, suggesting broad-spectrum EV and RV activity. Moreover, SERT/DAT/NET experiments suggest that the compounds, unlike (S)-fluoxetine, are not neuroactive.

Several structurally disparate 2C inhibitors have already been identified. With the exception of fluoxetine, dibucaine, pirlindole, and several other 2C inhibitors presented at a conference, the anti-enteroviral spectrum of most 2C-targeting compounds has not been systematically characterized [7, 22, 44]. The FDA-approved drugs fluoxetine (racemic mixture), dibucaine, and pirlindole and the parental compound 1 showed a limited antiviral spectrum and low potency [20]. We speculate that the broader antiviral spectrum of 2C inhibitors correlates with an increased antiviral activity and potency. Our previous finding that higher antiviral activity of (S)-fluoxetine resulted in a broader antiviral spectrum than the less active racemic mixture lends support to this hypothesis [24]. In our study, we identified several compounds that showed higher antiviral potency as well as a broader spectrum. The 4 compounds that showed the highest antiviral activity (i.e., 12a, 12b, 19b, and 19d) showed broad-spectrum EV and RV activity. These data lend further support to the proposed correlation between the antiviral potency of 2C inhibitors and the broadness of their antiviral spectrum.

To validate the broad-spectrum EV antiviral activity of 12a, 19b, and 19d, several clinical isolates of the serotypes EV-A71, CV-A24v, and EV-D68 were tested. Indeed, the compounds inhibited all clinical isolates, albeit with subtle differences. We reported previously that (S)-fluoxetine does not inhibit EV-A71 strain BrCr. Interestingly, several EV-A71 clinical isolates were inhibited by (S)-fluoxetine [20, 24]. The differences in the antiviral spectrum could be due to very minor intrinsic genetic differences between the clinical isolates of EV-A71. However, the observation that all isolates are inhibited by compounds 12a, 19b, and 19d but not the parental compound 1 makes it more likely that there is a correlation between antiviral potency and antiviral spectrum.

Combining the chemical moieties of 12a with 19b resulted in the most potent broad-spectrum EV and RV inhibitor. This compound, 12b, inhibited CV-B3, EV-D68, and HRV-B14 in the nM range, EV-A71 in the sub-µM range, and EV-C and HRV-A2 in the low µM range. The selectivity index, a parameter to express a compound's in vitro efficacy in the inhibition of virus replication, showed therapeutically relevant values in the range of 100 up to 3,000 and even higher against CV-B3 for the broad-spectrum EV and RV inhibitors (S3 Table). Together, these data show that 2C is an excellent target for highly potent antiviral drugs and that broad-spectrum 2C inhibitors can be developed in the therapeutically relevant nM range.

Raising compound-resistant viruses can reveal important insights into the mode of action of these compounds and their binding site. It was previously shown that mutation C179F conferred resistance to compound 1, whereas mutations I227V and A224V-I227V-A229V provided resistance to the thiazolobenzimidazol TBZE-029 and (S)-fluoxetine [20, 23, 24, 31]. We tested each of these mutations for their resistance toward compound 1, 12a, 19d, and (S)-fluoxetine. Compound 1 and 12 showed a similar resistance profile as (S)-fluoxetine. The resistance profile of 19d was slightly different. Unlike to the other compounds, mutations C179F and F190L provided some 19d resistance, but mutations I227V and A224V-I227V-A229V did not provide any resistance. These subtle differences in the resistant profiles of compound 1 and 12a on the one hand and 19d on the other hand may be explained by small differences in the interaction between the compounds and 2C, but the exact reason remains to be determined.

To gain more insight in how EVs can develop resistance, we raised 19d-resistant pools of EV-A71, CV-B3, and EV-D68. The importance of the mutations that were observed in these pools was evaluated by introducing them, alone or in combination, in recombinant viruses using reverse genetics. The recovered viruses were characterized for resistance as well as virus fitness. We observed a common mechanism in resistance development toward 19d in the serotypes EV-A71 and CV-B3. First, mutations in the α2 helix of 2C, or very close to it, were acquired that conferred 19d resistance but at the same time reduced viral fitness. Second, resistant EV-A71 and CV-B3 virus pools acquired additional mutations that are distant from the α2 helix. These distal mutations alone provided little if any resistance. But the combination of α2 helix mutations together with the distal mutations increased 19d resistance and additionally restored viral fitness. In addition, we raised 19d-resistant EV-D68 viruses. Similar to EV-A71 and CV-B3, a resistance mutation—T196S—arose close to the α2 helix. Interestingly, we also identified an EV-D68 virus pool that carried the mutation V96M. To our knowledge, this is the first mutation against a 2C inhibitor that is located outside the known structure. This suggests that the ATPase domain works in concert with the enigmatic N-terminal domain of 2C. We propose that mutations in, or in proximity to, the α2 helix of 2C disrupt compound binding, which results in loss of viral fitness. Additional distant mutations are required to compensate for the fitness defects and to further increase resistance.

2C is an SF3 helicase, which typically form hexameric assemblies. Such proteins have Walker A, B, and C motifs which form the catalytic site for ATPase activity. The energy produced by ATP hydrolysis induces a series of conformational changes that drive unwinding of RNA or DNA by SF3 helicases [45]. To support the hypothesis that the α2 helix of 2C forms part of the compound binding pocket, we predicted solvent accessible tunnels in the 2C proteins of EV-A71 and PV-1. We identified several tunnels that intersect at the α2 helix of EV-A71 2C. One of these tunnels was also present in the PV-1 2C. On basis of these observations, we propose that there is a common druggable binding pocket close to the α2 helix of 2C that is conserved in all the EVs and RVs. In support of this, 19d-resistant virus pools conferred cross-resistance to the broad-spectrum EV and RV inhibitors 12a and 19b. Additionally, CV-B3 and EV-D68 virus pools also conferred cross-resistance to (S)-fluoxetine. Furthermore, the compounds (S)-fluoxetine and 19d caused a dose-dependent shift in the melting temperature of recombinant CV-B3 2C protein suggestive of direct binding. Both compounds did not cause a shift in melting temperature of the F190L 2C mutant. This lends further support that the α2 helix is part of the binding pocket and that mutations in the α2 helix disrupt compound binding.

All predicted tunnels are very close to the Walker B domain but on the opposite side from where ATP is coordinated. This indicates that the compounds likely inhibit the 2C protein allosterically and do not directly occupy the ATP binding site (S2 Fig Panel E). Several possibilities for allosteric inhibition of the 2C protein can be proposed. Inhibitor binding could prevent the proper assembly of the functional 2C oligomer, or conversely, the inhibitors might stabilize the 2C protein and prevent conformational changes during ATP hydrolysis. This mechanism has already been shown for a small molecule inhibitor of the human AAA + ATPase, p97 [46]. The allosteric p97 inhibitor binds at the interface of two adjacent protein domains and thereby prevents propagation of the conformational changes that are necessary for proper p97 ATPase function. The lack of oligomeric 2C structures makes it difficult to elucidate whether the compounds would stabilize or destabilize the quaternary structure, or inhibit the enzymatic activities of 2C. Crystallographic or cryogenic electron microscopy structures of oligomeric 2C in complex with inhibitors are needed to clarify the binding site and the mode of action of 2C inhibitors.

In conclusion, we identified several novel, highly potent inhibitors with broad-spectrum EV and RV activity. Our data suggest that the 2C proteins of EV and RV share a common,

druggable binding pocket. Additionally, these compounds are less cytotoxic compared to the parental compound 1. The structure of the novel analogues resembles that of fluoxetine, but importantly, they lack the tri-fluoro moiety that is required for the SSRI activity of fluoxetine. This suggests that the novel inhibitors do not act as SSRIs, but this remains to be experimentally proven in vivo. Since the compounds are not FDA approved, pharmacological, pharmacokinetics, and toxicological data are required. Given the broad-spectrum activity of these novel compounds, we believe these to be promising candidates for further preclinical assessment. Eventually, these compounds could be developed into urgently needed broad-spectrum antivirals to combat EV and RV infections.

# Materials and methods

## Ethics statement

All experimental procedures involving animals were in accordance with Dutch law and approved by the Ethical Committee for Animal Experimentation of Utrecht University and the Central Committee Animal Experimentation (CCD; #AVD108002016443-1). Animals were treated humanely, and all efforts were made to minimize the number of animals used and their suffering.

## Cells

BGM cells (purchased from European Cell Culture Collection [ECACC]), Rhabdomyosarcoma (RD) cells (American Type Culture Collection [ATCC]) and HeLa R19 cells (ATCC) were cultured in Dulbecco's Modified Eagle Medium (DMEM; Lonza, Switzerland) supplemented with 10% (vol/vol) fetal calf serum (FCS; Lonza). HAP1 cells were obtained from Horizon Discovery Group plc (Cambridge, UK) and cultured in Iscove's Modified Dulbecco's Medium (IMDM; Lonza) containing 10% (vol/vol) FCS. Huh7-Lunet 7/T7, a stable cell pool expressing T7 RNA polymerase and blasticidin S-deaminase [47], were kindly provided by Volker Lohmann (Universitätsklinikum Heidelberg, Germany) and cultured in DMEM supplemented with 10% FCS and 10 μg/mL blasticidin (Sigma-Aldrich, Zwijndrecht, The Netherlands). All cell lines were grown at 37 ˚C in 5% $CO_2$. HEK293 cells expressing human DAT, NET, or SERT (kindly provided by Dr. Hoener from F. Hoffmann-La Roche Ltd., Basel, Switzerland) were cultured as described previously [42, 48]. Briefly, transfected HEK cells were cultured in T75 flasks (Thermo Fisher Scientific, Massachusetts) at 37 ˚C and 5% $CO_2$. DMEM high glucose (41965–039) was supplemented with 10% dialyzed fetal bovine serum (FBS), 2 mM L-glutamine, 1% 5,000 U/mL–5,000 μg/mL penicillin/streptomycin, 1 mM sodium pyruvate, 1% minimum essential medium non-essential amino acids solution (MEM-NEAA) solution, and 5 μL/mL geneticin selective antibiotic. Trypsin-EDTA (0.05%) was prepared by diluting trypsin-EDTA (0.5%) in phosphate-buffered saline (PBS). All cell culture materials were obtained from Gibco (Life Technologies, Breda, The Netherlands). Medium was refreshed every 2–4 days, and cells were passaged at >80% confluence with the use of PBS and trypsin-EDTA (0.05%) for up to 10 passages. The cell lines were routinely tested for mycoplasma contamination.

## Synthesized compounds and reagents

GuaHCl and (S)-fluoxetine were purchased from Sigma-Aldrich. GuaHCl was dissolved in water at 2 M stock concentration, and all other compounds were dissolved in DMSO at 10 mM stock concentration. Neurobasal-A medium, penicillin–streptomycin (5,000 U/mL–5,000 μg/mL), B27 plus supplement, N2 supplement, and l-glutamine are obtained from Life

Technologies (Bleiswijk, The Netherlands). Polyethyleneimine (PEI) solution (50%), sodium borate, boric acid, and all other chemicals (unless stated otherwise) are obtained from Sigma-Aldrich (Zwijndrecht, The Netherlands). General chemical synthesize route and NMR spectra of the compounds are listed in S1 Text.

## Viruses

EV-A71 (strain BrCr), PV-1 (strain Sabin, ATCC), and the EV-D68 strains Fermon, 431100074, 4310901348, 4310902042, 4310900947, and 4310902284 were obtained from the National Institute for Public Health and Environment (RIVM) in the Netherlands. HRV-2 and HRV-14 were obtained from Joachim Seipelt from the Medical University of Vienna in Austria. CV-B3 (strain Nancy) was obtained by transfecting BGM cells with RNA transcripts derived from the full-length infectious clones p53CB3/T7 as described by Wessels and colleagues [49]. EV-A71 clinical isolates TW/70811/08, TW/96016/08, TW/72232/04, TW/2639/04, and TW/2728/04 were kindly provided by Johan Neyts of the REGA Institute for Medical Research (KU Leuven, Belgium).

## Resistance selection and phenotyping of resistant virus variants

We raised compound 13–resistant EV-A71, CV-B3, and EV-D68 viruses as described earlier [30]. In short, the lowest concentration of 19d and the highest virus input that showed reproducible inhibition of viral CPE was determined. Next, three 96-well plates containing HeLa R19 cells were treated with the lowest determined compound concentration and the highest virus input to select for viruses outgrowing the compound. CPE development was monitored daily, and 3 days post infection, samples showing CPE were harvested after three freeze-thawing cycles. The harvested virus isolates were titrated in the presence of same concentration of 19d. After 3 days, the lysates of the highest virus dilution showing full CPE was harvested, and the isolates were expanded in a 25 cm$^2$ flask in the presence of 19d. In parallel, virus without compound was taken along and used as a reference. The virus titers from the obtained virus variants were determined by endpoint dilution, and the P2 and P3 regions were sequenced. The resistance and cross-resistance phenotype of the virus variants was determined with a multicycle viral replication assay.

## Reverse engineering of resistance virus variants

The CV-B3 mutations 2C[C179F], 2C[C179Y], 2C[F190L], 2C[I227V], and 2C[A224V/I227V/A229V] were previously introduced into the p53CB3/T7 infectious clone [24]. To obtain Rluc-CV-B3 reporter viruses, which contains the Renilla luciferase gene upstream of the capsid coding region, the 2C mutations were cloned into pRLuc-53CB3/T7 using BssHII (nt 4239) and XbaI (nt t4948) [50]. Mutations 2C[H243R] and 2C[F190L/H243R] were introduced into the pRLuc-53CB3/T7 backbone with the Q5 site-directed mutagenesis kit (New England Biolabs, Bioké, Leiden, The Netherlands). After site-directed mutagenesis, the plasmids were subjected to Sanger sequencing to ensure the existence of the introduced mutation. Rluc-CV-B3 wild-type and 2C mutant viruses were obtained by transfection of MluI-linearized plasmid DNA into the Huh7-Lunet 7/T7 cells as described earlier [47, 50].

The EV-A71 mutations 2C[T133S], 2C[R151H], 2C[M193L], 2C[M193L/R151H], and 2C[M193L/R151H/T133S] and the EV-D68 mutations 2C[V96M] and 2C[T196S] were introduced into the pEV-A71 backbone (kindly provided by Johan Neyts) [51] and pRib-EVD68 Fermon [52], respectively, using the Q5 site-directed mutagenesis kit (New England Biolabs, Bioké, Leiden, The Netherlands). After site-directed mutagenesis, the plasmids were subjected to Sanger sequencing to ensure the existence of the introduced mutation. The corresponding

primers used for side-directed mutagenesis of the aforementioned viruses are reported in S4 Table. To obtain virus, the plasmids were linearized with MluI, and RNA was in vitro transcribed using the T7 RiboMAX Express Large Scale RNA production system (Promega, Leiden, The Netherlands) according to the manufacturer's protocol and transfected into HeLa R19 and RD cells. To ensure that the introduced mutations were retained in the generated virus, viral RNA was isolated with the NucleoSpin RNA Virus kit (Macherey-Nagel, Leiden, The Netherlands) according to the manufacturer's protocol, and the presence of the desired mutations was confirmed by Sanger sequencing. Virus titers were determined by endpoint dilution titration and calculated according to the method of Reed and Muench and expressed as 50% cell culture infective dose ($CCID_{50}$) [53].

## Single-cycle virus infection

Virus infections were performed by incubating subconfluent HeLa R19 cells with virus at the indicated MOI at 37 ˚C for 30 minutes. Next, the inoculum was removed, and fresh control medium or compound-containing medium was added to the cells. At the indicated time points, cells were frozen. For measurements of infectious particles, virus was released from the cells by three freeze-thawing cycles. Virus titers were determined by endpoint dilution assay and calculated by the method of Reed and Muench and expressed as 50% tissue culture infectious dose ($TCID_{50}$) [53]. In the case of RLuc-CV-B3 infection, cells were lysed at the indicated time points post infection, and the *Renilla* luciferase Assay System (Promega, Leiden, The Netherlands) was used to determine the luciferase activity. Cell viability was determined in parallel using the AQueous One Solution Cell Proliferation Assay (Promega, Leiden, The Netherlands) according to the manufacturer's protocol. Optical density at 490 nm was determined using a microplate reader.

## Multicycle virus infection

Subconfluent layers of HeLa R19 cells were seeded in 96 wells and treated with serial dilutions of the corresponding compounds. Cells were infected with viruses at the lowest possible MOI resulting in full CPE within 3 days (MOI 0.001 for EV-A71, EV-A71 clinical isolates, CV-B3, and PV-1; MOI 0.01 for CV-A24, CV-A24v, and HRV-14; MOI 0.1 for EV-D68, EV-D68 clinical isolates, and HRV-2). Subsequently the cells were incubated at 37 ˚C for 3 days until full CPE was observed in the virus-infected untreated cell controls. Cell viability was determined using the AQueous One Solution Cell Proliferation Assay (Promega, Leiden, The Netherlands) according to the manufacturer's protocol. The optical density at 490 nm was determined using a microplate reader. Raw OD values were converted to percentage of untreated and uninfected cell control after subtraction of the background. The concentration of compound that inhibits virus-induced cell death or expression of *Renilla* luciferase by 50% ($EC_{50}$) was calculated by nonlinear regression analysis with GraphPad Prism Version 6. We set the threshold for compound activity at concentration of 10 μM or less since low μM range is typical for antivirals in clinical use. Cytotoxicity of the compounds was assessed in a similar set-up, and $CC_{50}$ values were derived from cell viability values determined with an MTS assay.

## Thermal shift assays

The DNA fragment encoding residues 116 to 329 of CV-B3 2C (strain Nancy) was cloned into a pET28b plasmid, downstream of an N-terminal, and 3C protease cleavable, hexahistidine-MBP tag. The F190L mutation was introduced using a Q5 site-directed mutagenesis kit (New England Biolabs, Bioké, Leiden, The Netherlands). The recombinant WT protein, and F190L variant, were produced in *Escherichia coli* RosettaTM 2 (DE3) (Sigma-Aldrich, Zwijndrecht,

The Netherlands). When cultures reached OD600nm of 0.5, protein expression was induced by the addition of 0.5 mM IPTG. Subsequently, protein was expressed for 16 h at 18 ˚C, with shaking at 200 rpm. Further protein purification steps were performed essentially as described previously [27], with the exception that TEV protease was replaced with 3C protease (Sigma-Aldrich). The final size-exclusion chromatography step was performed at 4 ˚C with buffer containing 25 mM Tris (pH 8), 300 mM NaCl, and 1 mM MgCl2, using a superose 6 increase 10/300 column GL (GE Healthcare Life Science, Eindhoven, The Netherlands). The binding of (S)-and (R)-fluoxetine and both 2b-enantiomers to WT CV-B3 2C, and the F190L variant, was monitored by the fluorescence-based thermal shift assay using a Roche LightCycler480, as described previously [27]. Each well of the thermal shift assay plates contained WT or F190L 2C protein (final concentration of 10 μM in 50 mM Tris, 300 mM NaCl, and 1 mM MgCl2 [pH 8]) and a SYPRO orange solution in concentrations recommended by the manufacturer, in a final volume of 25 μL. All thermal shift assay experiments were performed using a temperature gradient ranging from 20 ˚C to 90 ˚C (incremental steps of 0.2 ˚C/12 s). Protein denaturation was monitored by following the increase of the fluorescence emitted by SYPRO orange, which binds to exposed hydrophobic regions of the denatured protein. The mid-log of the transition phase from the native to the denatured protein was used to calculate the melting temperature (Tm). The reference unfolding temperature of proteins in 5% DMSO (T0) was subtracted from the values in the presence of compounds (Tm) to obtain thermal shifts, $\Delta Tm = (Tm - T0)$.

## Rat neuronal cell culture preparation for recording of neuronal activity

Primary rat cortical cells were isolated from PND0-1 Wistar rat pups as described previously [54, 55]. Briefly, PND0-1 pups are decapitated, and cortices were rapidly dissected on ice and were kept in serum-free dissection medium (Neurobasal-A supplemented with 25 g/L sucrose, 450 μM l-glutamine, 30 μM glutamate, 1% penicillin/streptomycin, and 2% B27 plus supplement [pH 7.4]) during the entire procedure. Cortices were dissociated to a single-cell suspension by mincing with scissors, trituration, and filtering through a 100 μm mesh (EASYstrainer, Greiner Bio-one, Solingen, Germany). The cell suspension was diluted to a $2 \times 10^6$ cells/mL solution, after which droplets of 50 μL were placed on the electrode fields in wells of pre-coated 48-well MEA plates (Axion BioSystems, Atlanta, GA). All cell culture surfaces were pre-coated with 0.1% PEI solution diluted in borate buffer (24 mM sodium borate/50 mM boric acid in Milli-Q adjusted to pH 8.4). After plating, cells were left to adhere for approximately 2 hours before adding 450 μL serum-free dissection medium. Rat primary cortical cells were maintained at 37 ˚C in a humidified 5% $CO_2$ incubator. At DIV4, 90% of the serum-free dissection medium was replaced with serum-free culture medium (Neurobasal-A supplemented with 25 g/L sucrose, 450 μM l-glutamine, 1% penicillin/streptomycin, and 2% B27 plus supplement [pH 7.4]). Rat primary cortical neurons were used for neurotoxicity assessment at DIV15.

## MEA recordings of spontaneous neuronal activity in rat primary cortical cultures

Each well of a 48-well MEA plate contains 16 nanotextured gold micro-electrodes (approximately 40–50 μm diameter; 350 μm spacing) with 4 integrated ground electrodes, yielding a total of 768 channels for simultaneous recording (for review see [43]). Spontaneous electrical activity was recorded as described previously [54, 55] (S3 Fig). Briefly, signals were recorded at the day of experiments (DIV15) using a Maestro 768-channel amplifier with integrated heating system, temperature controller, and a data acquisition interface (Axion BioSystems, Atlanta, GA). Data acquisition was managed with Axion's Integrated Studio (AxIS version 2.4.2.13)

and recorded as .RAW files. All channels were sampled simultaneously with a gain of 1,200× and a sampling frequency of 12.5 kHz/channel, using a 200–5,000 Hz band-pass filter.

Prior to the recording, MEA plates were allowed to equilibrate for 5–10 minutes in the Maestro. Following a 30-minute baseline recording, wells were exposed to the test compounds or the DMSO control, and activity was recorded for another 30 minutes to determine the acute effects of test compounds on neuronal activity. Stock solutions of compounds dissolved in DMSO (10 mM) were diluted (1:100) in serum-free culture medium to obtain 100 μM solutions, which were used for exposure of cells in the MEA (dilution 1:10; final concentration 10 μM/0.1% DMSO). In order to prevent receptor (de)sensitization, each well was exposed only to a single concentration.

To determine (modulation of) spontaneous activity, RAW data files were re-recorded to obtain Alpha Map files for further data analysis. In this re-recording, spikes were detected using the AxIS spike detector (Adaptive threshold crossing, Ada BandFIt version 2) with a variable threshold spike detector set at 7× SD of internal noise level (rms) on each electrode. Post/ pre-spike duration was set to 3.6/2.4 ms. For further data analysis, spike files were loaded in NeuralMetric Tool (version 2.2.4, Axion BioSystems). Only active electrodes (MSR $\geq$ 6 spikes/ min) in active wells ($\geq$1 active electrode) were included in the data analysis. The (network) bursting behavior was analyzed using the Poisson Surprise method [56] with a minimal surprise of 10 and a minimum bursting frequency of 0.3 bursts/min. Network bursts were extracted with the adaptive threshold algorithm. Effects of test compounds on the spontaneous activity pattern were determined by comparing activity during exposure to baseline activity. To prevent inclusion of exposure artefacts, the window of 20–30 minutes post exposure was used for analysis of effects. A custom-made Microsoft Excel macro was used to calculate treatment ratios (TRs) per well for the different metric parameters (MSR, MBR, and MNBR) by: (parameter$_{exposure}$/parameter$_{baseline}$) × 100%. Hereafter, TRs will be normalized to DMSO control. Wells that show effects 2× SD above or below average were considered outliers (5.1%) and were removed for further data analysis.

## Inhibition of uptake by monoamine transporters

Uptake activity of hNET, hDAT, and hSERT was measured using the Neurotransmitter Transporter Uptake Assay Kit from MDS Analytical Technologies (Sunnyvale, CA) as described previously [42, 48]. Briefly, the kit contained a mix consisting of a fluorescent substrate, which resembles the biogenic amine neurotransmitters, and a masking dye that extinguishes extracellular fluorescence. The fluorescent substrate solution was prepared by dissolving the mix in HBSS according to the manufacturer's instructions and stored at −18 ˚C for a maximum of 4 days. Uptake of the fluorescent substrate increases intracellular fluorescence, while extracellular fluorescence is blocked by the masking dye (Fig 8) [57].

On day 0, HEK 293 cells were seeded at a density of approximately 60,000 cells/well in clear-bottom, black-walled, 96-well plates (Greiner Bio-one, Solingen Germany) coated with PLL buffer (50 mg/L). Cells were allowed to proliferate overnight in a humidified 5% CO$_2$/95% air atmosphere at 37 ˚C. Experiments were performed the next day (day 1). Cells were pre-incubated with the fluorescent substrate for 12 minutes (t = −12 to t = 0) prior to a 30-minute drug exposure. Culture medium was replaced by 100 μL/well fluorescent substrate solution, and uptake measurements were started. At t = 0, 100 μL/well HBSS with 0.1% DMSO (control) or with test compound (final concentration 1–100 μM) was added to each well, and uptake was measured continuously for 30 minutes. Background wells were pre-incubated with 100 μL/well HBSS without fluorescent substrate solution and exposed at t = 0 min to 100 μL/ well HBSS.

Fluorescence was recorded every 3 minutes, starting directly after addition of the fluorescent substrate solution (t = −12) using a microplate reader (Tecan Infinite M200 microplate; Tecan Trading Männedorf, Switzerland) at 37 ˚C at 430/515 nm excitation/emission wavelength in bottom-reading mode using optimal gain values for each cell type (number of cycles: 21; time interval: 3 minutes; number of flashes: 19; integration time: 20 μs, no lid). Cell attachment was visually examined following experiments.

The fluorescence of each well was background corrected (time- and plate-matched). First, uptake of the fluorescent substrate was first determined per well by calculating the change in fluorescence (ΔFU) at 12 minutes after exposure to the test compound (t = 12) compared to the fluorescence prior to exposure (i.e., the fluorescence following 12-minute pre-incubation with the fluorescent substrate at t = 0), as a percentage of the fluorescence prior to exposure. Second, the percentage uptake in control wells of all plates was averaged, and wells that showed values 2× SD above or below average were considered outliers and were excluded from further analysis. Uptake in compound-exposed wells was expressed as a percentage of control wells, outliers in exposed groups (effects 2× SD above or below average) were removed (approximately 1%), and all uptake values were scaled between 0% and 100%.

### Prediction of solvent accessible tunnels

For prediction of solvent accessible tunnels, we used the crystal structure of EV-A71 2C PDB: 5GRB/Chain B and PV 2C PDB 5Z3Q/Chain A. As a starting point for the tunnel prediction, the amino acid of 2C of EV-A71 M193 and PV-1 2C F164—N179 and M187, respectively—were used. We used the MOLEonline web interface [40], CAVER 3.0PyMol plugin [58], and CASTp [59] to generate solvent accessible tunnels or surfaces [40, 58, 59]. Figures were generated using UCSF Chimera and PyMol (The PYMOL Molecular Graphics system Version 234932948 Schrödinger, LLC).

### Statistical analysis

Each experiment was performed at least in technical duplicates, and all experiments were performed in biological triplicates. Statistical significance of the MEA data and transporter inhibition was determined using one-way ANOVA and, if applicable, a post hoc Tukey test. $P \leq 0.05$ was considered significant. Statistical analysis, nonlinear regression, and the graphs were made with GraphPad Prism Version 6 or R version 3.6.0 (R core team 2019).

### Supporting information

**S1 Text. Detailed description of synthesis of compound 1 analogues and corresponding NMR spectra.**
(DOCX)

**S1 Data. Raw experimental data of Tables 1–4 and S1–S3 Tables.**
(XLSX)

**S2 Data. Raw experimental data of Figs 2–8 and S4 Fig.**
(XLSX)

**S1 Table. Antiviral activity of the furan amide moiety analogues against CV-B3.** Multicycle viral replication assays were performed in HeLa R19 cells, and shown are $EC_{50}$ and $CC_{50}$ values in μM. Data represent mean ± SD calculated from two different experiments both performed in biological triplicates. All underlying experimental data are displayed in S1 Data.
(DOCX)

**S2 Table. Antiviral activity of the N-benzylaniline moiety analogues against CV-B3.** Multicycle viral replication assays were performed in HeLa R19 cells, and shown are $EC_{50}$ and $CC_{50}$ values in μM. Data represents mean ± SD calculated from two different experiments both performed in biological triplicates. All underlying experimental data are displayed in S1 Data. (DOCX)

**S3 Table. Antiviral efficacy against CV-B3 and cytotoxicity of the active compounds in different cell lines.** Multicycle viral replication assay was performed in HeLa R19, HAP1, and BGM cells. Shown are $EC_{50}$ and $CC_{50}$ values in μM. Data represent mean ± SD calculated from two independent experiments both performed in biological triplicates. SI = selectivity index (CC50/EC50). All underlying experimental data are displayed in S1 Data. (XLSX)

**S4 Table. Primers used for site-directed mutagenesis.** Primers used for introducing mutations in CV-B3, EV-A71, and EV-D68. Mutations introduced are shown in small characters in the primer sequence. (XLSX)

**S1 Fig. Synthesis routes of compound 1 analogues to explore the furan ring and the amide bond.** Synthetic route a): (i) NaBH4, MeOH/THF (4:1), rt, 6 h, quantitative; (ii) Compounds 5a–5d: corresponding acyl chloride or sulphonyl chloride, TEA, DCM, rt, 3h, 63%–98%; Compounds 1, 5e–f: corrisponding carboxylic acids, TBTU, DIPEA, DMF, rt, on, 37%–69%; Compounds 5g: Na(AcO)$_3$BH, MeOH, rt, on, 30%; (iii) 2-bromoacetyl chloride, TEA, DCM, rt, 1h, 68%; (iv) methylamine, EtOH, rt, on, 66%; Synthetic route b): (i) K$_2$CO$_3$, benzyl bromide, DMF, rt, 18h 68%; (ii) Boc, DMAP, TEA, THF, rt, 18h, 90%; (iii) Pd/C 10%, H$_2$ atmosphere, EtOAc/MeOH (1:1, rt, 20h, 93%; (iv) compound 4 or compound 18b, TBTU, DIPEA, DMF, 45 ˚C, 48–72h, 79%–45%; (v) DCM/TFA (1:1), rt, 4h, 91%–89%. Synthetic route c): (i) NaBH4, MeOH/THF (4:1), rt, 6 h, 88%; (ii) 4-fluorobenzoyl chloride, TEA, DCM, rt, 3h, quantitative. (TIF)

**S2 Fig. Synthesis routes of compound 1 analogues to explore the 4-positions of the _N_-benzyl aniline moiety.** (i) NaBH4, MeOH/THF (4:1), rt, 6 h, 40%–99%; (ii) furan-2-carbonyl chloride, TEA, DCM, rt, 3 h, Y = 56%–99%; (iii) thionyl chloride, DCM, reflux, 2h; (iv) compound 18d, TEA, DCM, rt, 3h, 75%. (TIF)

**S3 Fig. Structure-based multiple sequence alignment of 2C proteins.** Multiple sequence alignment of EV-A71 (strain BrCr), CV-B3 (strain Nancy), PV (strain Sabin), and EV-D68 (strain Fermon) was performed with Clustal Omega. Invariant amino acids are highlighted in red. Secondary structural elements are shown on top of the alignment and are based on the EV-A71 crystal structure (PDB: 5GRB). Functional motifs are indicated in black. The green box indicates resistance mutations which are located at or close by the α2 helix. The yellow boxes highlight distal mutations. (TIF)

**S4 Fig. In silico prediction of a possible binding pocket around the α2 helix of 2C.** The mole online version (40) was used to calculate solvent exposed tunnels in 2C of PV-1 (PDB: 5z3Q) from different starting points. (A) The starting point for calculation was the amino acid F164, and in (B) the starting point for the tunnel prediction was N179. The Pymol plug-in CAVER 3.0.3 was used to calculate solvent accessible tunnel in the nonstructural protein 2C (58). (A) The EV-A71 2C crystal structure PDB: 5GRB, chain B was used. The amino acid

M193 in the α2 helix of 2C is depicted in green and represents the starting point to identify solvent accessible tunnels. The identified tunnels are shown red, blue, green, and yellow. (B) For PV, the 2C crystal structure 5Z3Q, chain B was used to identify solvent accessible tunnels using the amino acid M187 as starting point. The identified tunnels are depicted in red, green, and blue. The CASTp online tool was used to predict cavities on the protein surfaces of the 2C nonstructural proteins of (C) EV-A71 and (D) PV (59). The cavities are highlighted in blue. (TIF)

**S5 Fig. Overview of the microelectrode array.** (A) Schematic overview of the microelectrode array (MEA) recordings used to measure changes in neuronal activity. Photographs of the Maestro 768-channel amplifier (A) and 48-well MEA plate (B). Each well contains 16 nanotextured gold micro-electrodes on top of which neuronal cells can be cultures for recording of spontaneous electrical activity (C). Baseline activity recorded before exposure (D, left) is compared to activity following exposure to a (inhibitory) test compound (D, right) to determine a TR that describes the changes in neuronal activity due to exposure to the test compounds. Modified after Tukker and colleagues, 2016 [55]. TR, treatment ratio. (TIF)

**S6 Fig. Modulation of spontaneous neuronal activity anti-EV inhibitors.** (A) Modulation of spontaneous neuronal activity by fluoxetine. Concentration–response curves for inhibition of MSR (left), MBR (middle), and MNBR (right) following acute exposure to fluoxetine. Neuronal activity is depicted as the mean TR ± SEM as percent of control (DMSO) wells ($n$ = 12–16 wells, derived from 2 independent cultures, $^{*}P < 0.05$). (B) Modulation of spontaneous neuronal activity by fluoxetine and antiviral compounds. When tested at a single, high concentration (10 µM), fluoxetine induced a profound inhibition of MSR (left), MBR (middle), and MNBR (right), whereas the antiviral compounds 12a, 12b, 19b, and 19d were without effect. Neuronal activity is depicted as the mean TR ± SEM as percent of control (DMSO) wells ($n$ = 8–16 wells, derived from 2 independent cultures, $^{*}P < 0.05$). All underlying experimental data that are displayed can be found in S2 Data. TR, treatment ratio. (TIF)

## Acknowledgments

We thank Prof. Friedrich Förster for providing access to equipment used for purification of recombinant 2C proteins.

## Author Contributions

**Conceptualization:** Lisa Bauer, Roberto Manganaro, Birgit Zonsics, Daniel L. Hurdiss, Salvatore Ferla, Johan Neyts, Jeroen R. P. M. Strating, Remco H. S. Westerink, Andrea Brancale, Frank J. M. van Kuppeveld.

**Data curation:** Lisa Bauer, Roberto Manganaro, Birgit Zonsics, Daniel L. Hurdiss, Marleen Zwaagstra, Tim Donselaar, Naemi G. E. Welter, Regina G. D. M. van Kleef, Moira Lorenzo Lopez, Federica Bevilacqua, Thamidur Raman, Salvatore Ferla, Marcella Bassetto, Remco H. S. Westerink.

**Formal analysis:** Lisa Bauer, Roberto Manganaro, Birgit Zonsics, Daniel L. Hurdiss, Naemi G. E. Welter, Regina G. D. M. van Kleef, Remco H. S. Westerink.

**Funding acquisition:** Daniel L. Hurdiss, Jeroen R. P. M. Strating, Andrea Brancale, Frank J. M. van Kuppeveld.

**Investigation:** Lisa Bauer, Roberto Manganaro, Birgit Zonsics, Daniel L. Hurdiss, Marleen Zwaagstra, Tim Donselaar, Naemi G. E. Welter, Regina G. D. M. van Kleef, Moira Lorenzo Lopez, Federica Bevilacqua, Thamidur Raman, Salvatore Ferla, Marcella Bassetto, Andrea Brancale, Frank J. M. van Kuppeveld.

**Methodology:** Remco H. S. Westerink, Andrea Brancale, Frank J. M. van Kuppeveld.

**Supervision:** Jeroen R. P. M. Strating, Remco H. S. Westerink, Andrea Brancale, Frank J. M. van Kuppeveld.

**Validation:** Lisa Bauer, Daniel L. Hurdiss, Andrea Brancale, Frank J. M. van Kuppeveld.

**Visualization:** Lisa Bauer, Daniel L. Hurdiss.

**Writing – original draft:** Lisa Bauer, Roberto Manganaro, Birgit Zonsics, Daniel L. Hurdiss, Salvatore Ferla, Remco H. S. Westerink, Andrea Brancale, Frank J. M. van Kuppeveld.

**Writing – review & editing:** Lisa Bauer, Roberto Manganaro, Birgit Zonsics, Daniel L. Hurdiss, Salvatore Ferla, Johan Neyts, Remco H. S. Westerink, Andrea Brancale, Frank J. M. van Kuppeveld.

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
