## [Editor Report · Decision Letter 0]

22 May 2020

Dear Dr van Kuppeveld, 

Thank you for submitting your manuscript entitled "Rational design of highly potent pan-enterovirus inhibitors targeting the non-structural protein 2C" for consideration as a Research Article by PLOS Biology.

Your manuscript has now been evaluated by the PLOS Biology editorial staff as well as by an academic editor with relevant expertise and I am writing to let you know that we would like to send your submission out for external peer review.

Please re-submit your manuscript within two working days, i.e. by May 24 2020 11:59PM.

Kind regards,

Di Jiang, PhD

Associate Editor

PLOS Biology

---

## [Decision Letter · Decision Letter 1]

24 Jun 2020

Dear Dr van Kuppeveld,

Thank you for sending us the new information in your appeal to our previous decision on your manuscript "Rational design of highly potent pan-enterovirus inhibitors targeting the non-structural protein 2C" which was evaluated by the PLOS Biology editors, an Academic Editor, and by three independent reviewers.

As you know that we have rescinded our last decision, and we now welcome re-submission of a much-revised version that takes into account the reviewers' comments AND includes the new compound with significantly improved potency compared with that in the version of the manuscript reviewed by our external reviewers. We will need to see that our reviewers are satisfied with the potency of the new derivative, and you will need to address all other concerns raised by the reviewers except the two mechanistic aspects that we overruled on and the exploration of SSRI activity which we overruled in our last decision. We cannot make any decision about publication until we have seen the revised manuscript and your response to the reviewers' comments. Your revised manuscript is also likely to be sent for further evaluation by the reviewers.

We expect to receive your revised manuscript within 2 months. 

**IMPORTANT - SUBMITTING YOUR REVISION**

*Re-submission Checklist*

*Published Peer Review*

*PLOS Data Policy*

*Blot and Gel Data Policy*

Sincerely,

Di Jiang, PhD

Senior Editor

PLOS Biology

REVIEWS:

Reviewer #1: Picornaviruses are significant human pathogens for which there is no vaccine or antivirals. This manuscript described the lead optimization based on compound 1, which was identified through a HTS. Several analogs including 12, 19b and 19d, were identified to have broad-spectrum antiviral activity against EV-A71, EV-D68, CVA24, Polio and HRV. The mechanism of action was mainly studied by the resistance-selection experiment. Additional evidence was provided by the thermal shift binding assay. The highlights of this study include:

1) Broad-spectrum antiviral profiling against multiple enteroviruses. 

2) Detailed pharmacological characterization. Using compound 19d as a chemical probe, resistant mutants have been selected in CV-B3, EV-A71 and EV-D68 viruses. Mutants were identified in the 2C protein, and the corresponding mutant viruses were generated through reverse genetics. 

Although the leads generated from this study are promising drug candidates for picornavirus antiviral, there are several issues that remain to be addressed:

1) The antiviral activity of the optimized hits 12,19b and 19d are on par similar to the (S)-fluoxetine (Table 5), although it is noted that the antiviral spectrum is expanded. The potency needs to be further optimized down to sub-micromolar to have clinical relevance. There are several series of 2C inhibitors in the literature that have submicromolar efficacy. Therefore, the results in this study are not particularly significant. 

2) The detailed mechanism of action at the molecular level remains elusive. Although resistant mutants have been identified, no docking models were shown to illustrate how compound 19d interact with the 2C protein. The results shown in Fig. 6 have no direct relationship with the drug binding. Overall the detailed mechanism of action remains a mystery and this study does not significantly advance our knowledge on the pharmacology of 2C inhibitors. 

3) The authors did not explain from the structural perspective why compounds 12, 19b and 19d have a broader spectrum of antiviral activity compared to fluoxetine. 

4) The overall approach is not particularly novel, it represents a standard medicinal optimization approach. Therefore this manuscript is more suitable for chemistry journals. 

Specific comments are:

1) It was claimed that one of the advantages of current hits is that they are not neuroactive. I am not sure what this actually means. Does this have to do with the potential side effects of fluoxetine analogs? The compounds described here are still at the early stage of drug development, and there are many parameters need to be optimized including microsomal stability, solubility, permeability, CYP inhibition and etc. Therefore neurotoxicity side effect is not a major concern at this stage. 

2) Line 113, "CPE-reduction assays" should read "CPE assay". There is only plaque reduction assay, not CPE reduction assay. This should be changed throughout the manuscript.

3) In table 2, the structures for R1 substitution are missing.

4) In Fig. 7, why the other two mutants H243R, F190L+H243R have not been tested? 

Reviewer #2: The authors report the identification of antiviral compounds with pan-enterovirus and pan-rhinovirus activity starting from a previously reported compound 1. The escape mutant analysis has permitted the authors to suggest that enteroviruses and rhinoviruses could share a druggable pocket in the 2C protein that needs further validation.

Major comments:

Overall this is a difficult paper to read and fully appreciate the key finding proposed by the authors that there is a common druggable pocket in entero- and rhinoviruses that can be targeted to try and obtain potent drugs. An alignment of 2C protein from the various strains examined in this study in relation to previous mutations in CV-B3 virus and the strains compared in this study may be useful for non-EV readers to try and integrate the findings in the different viruses. Data on SSRI activity of the new compounds could add value to attractiveness of this compound which presumably needs to be further optimized for potency.

Minor comments:

Table 2 is incomplete.

Line 42 : …numbered enteroviruses and rhinoviruses? 

Line 335: … should it be "mutants were not tested"

Line 714: … should it be "end-point dilution assay"

Reviewer #3: Enteroviruses are ubiquitous pathogens. Despite decades of trying, no direct acting antiviral medications exist. This is an extremely well written paper that describes the exploration of anti-enteroviral activities and cytotoxicity of derivatives of a compound described originally by Zuo et al (ref 20). The authors, noting the structural similarities between this compound and fluoxetine (shown by Dr. van Kuppeveld et al and Zuo et al to inhibit replication) carefully modified key elements of this molecule and in the process identified congeners with broader antiviral activity, reduced impact on neuronal activity and lower cytotoxicity. The paper closes somewhat speculatively by analyzing models of the target of action of these compounds, the viral 2C protein. 

The paper is highly detailed and may have initially been composed with the goal of publishing it in a medicinal chemistry journal. The detailed description of the schema for synthesis does not detract from the paper, and instead makes it all the more impressive. The only criticisms of this detail may be the complexity of the figure legends, and several small errors in labeling. For example, in line 163 the text refers to "final product 9", which appears to refer to compound 15 (the final product of Synthesis route C). Also, in two places "13d" (e.g. line 347) is mentioned, when 19d is intended. 

Several other minor errors are present in Table legends for Table 4 and 5 with begin wiht "Multicycle (sic) was performed in Hela R19 cells,.." (Multicycle replication assay were performed..." may have been intended).

Also, a number of terms and abbreviation may need definition for the audience of this journal.

Two somewhat important points that should be addressed follow:

First, the authors should define levels of antiviral activity. Most experts would not consider EC50 values of 5 to 10 micromolar to be more than mild to moderate; submicromolar is more typical of molecules that come to be used therapeutically. The paper would be improved by mentioning this or similar stratification.

One related point: the EC-50 values for compound 12 CV-B3 (Nancy) of ~10 to 20 nM are very impressive, but since this strain no longer circulates, one wonders why other enterovirus B virus strains were not examined, as was done with EV-A71, EV-D68, and CVA24. 

Addressing these two points would not detract from this excellent paper, but as it is, the authors seem a bit overly enthusiastic in their characterization of their best compounds as candidates for clinical development.

---

## [Decision Letter · Decision Letter 2]

19 Aug 2020

Dear Frank,

Thank you very much for submitting your revised manuscript "Rational design of highly potent pan-enterovirus inhibitors targeting the non-structural protein 2C" for consideration as a Research Article by PLOS Biology and please accept my apologies for the time it has taken us to return to you with a decision on the work. Your work has now been seen by reviewers 1 and 3, whose comments you will find at the end of this email, an academic editor and the staff editorial team.

Based on the reviews, I am delighted to say that we can in principle offer to publish the work, conditional on addressing a few outstanding referee and editorial (reporting and formatting) requests. Please address the issues raised by reviewer 3, which are self-explanatory. 

Before we will be able to formally accept your manuscript and consider it "in press", we also need to ensure that your article conforms to our guidelines. A member of our team will be in touch shortly with a set of requests. In addition while checking through the manuscript and associated files, we noticed the following editorial points which we will need you to address. As we can't proceed until all these requirements are met, your careful attention to addressing all issues (the below and the ones sent by my colleague in a subsequent email) will help prevent delays to publication. 

1. As indicated by reviewer 1 (opinion shared by us and our Academic Editor), the manuscript seems currently geared to an organic chemistry/chemical biology audience. As our readership is eminently composed of biologists, the manuscript will need to be thoroughly reworked with this in mind (without removing data). As part of these efforts, and in an attempt to make the work more accessible for a life science readership, we would strongly suggest to move Schemes 1 and 2 (which should be relabelled as two panels within a figure) and tables 2, 3 into the supplementary information.

2. We would suggest a title that more accurately reflects the manuscript, such as "Rational design of highly-potent broad-spectrum enterovirus inhibitors targeting the non-structural protein 2C", in addition to addressing reviewer 3s point about the pan-inhibition claimed in the manuscript text.

3. Please note that there are currently two Table 1 (on in the Methods), and relabelling will be needed, accordingly. The Methods table could also be included in the supplement.

4. Please include an "Ethics Statement", labelled as such, as a first subheading in the Methods section. This should also include the protocol number whereby your Ethical Committee granted approval of these experiments (in addition to the other relevant informaiton you currently have elsewhere in the Methods). 

5. The "Calculations and Graphs" section should be relabelled "Statistical Analyses" and include all of the statistical analysis information that is currently scattered in other subsections in the Methods. The information in the figure legends should not change (i.e. the statistical information should also be there).

6. Please nclude a "Cells" subsection detailing the provenance (commercial or otherwise) of every cell line used, whether they have been authenticated in your laboratory (if so, by which method) and whether they have been tested for mycoplasma contamination.

7. We will need a written statement of whether the photos in SF3 have been generated by you or have been published elsewhere. If so, we can only use them if they have been published under a CC-BY license.

8. You may be aware of the PLOS Data Policy, which requires that all data be made available without restriction: http://journals.plos.org/plosbiology/s/data-availability.

All individual quantitative observations that underlie the data summarized in the figures and results of your paper need to be made available. Thus, please provide with your final manuscript supplementary files (e.g., excel), uploaded as 'Supporting Information' and referred to (in the manuscript, figure legends, and the Description field when uploading your files) using the following format: S1 Data, S2 Data, etc. 

Multiple panels of a single or even several figures can be included as multiple sheets in one excel file that is saved using exactly the following convention: S1_Data.xlsx (using an underscore). The Excel files must be clearly labelled such that the experiments and data points they refer to are easy to understand. 

Your manuscript needs underlying data for the following figures: Fig 2, Fig 3, Fig 4 B, C, E, F, H, I, Fig 6, 7B, 8B, SF4. Each of those figure legends needs to state where the underlying data can be found (e.g. Source data for panel X can be fund in S1 Data, as applicable).

We expect to receive your revised manuscript within two weeks. Your revisions should address the specific points made by reviewer 3 and the editorial points above. 

*Submitting Your Revision*

To submit your revision, please go to https://www.editorialmanager.com/pbiology/ and log in as an Author. Click the link labelled 'Submissions Needing Revision' to find your submission record. Your revised submission must include:

- a cover letter detailing the changes made in response to the editorial requests made in this letter

- a Response to Reviewers file that provides a detailed response to any reviewers' comments, if applicable 

ADDITIONAL INFORMATION

*Copyediting*

*Published Peer Review History*

*Early Version*

With best wishes,

Nonia

Nonia Pariente, PhD,

Senior Editor,

djiang@plos.org,

PLOS Biology

Reviewer remarks:

Reviewer #1: During the revision, the authors made an effort to develop an additional compound 12b that improved antiviral activity. The highlights of this manuscript include:

1) A lead compound 12b with broad-spectrum antiviral activity against enteroviruses and rhinoviruses were identified. 

2) The mechanism of action was studies by resistance selection using CVB3, EV-A71, EV-D68 viruses. Drug resistance was confirmed by reverse genetics and thermal shift binding assay.

3) Mutations were mapped to the alpha2 helix and solvent accessible tunnels were identified in this region.

Weaknesses of this manuscript include:

1) Compared to the previously publications related to 2C inhibitors, this current study did not significantly advance our knowledge how 2C inhibitors interact with the drug target. The mutations for the current compounds were mapped to the same region as previously reported 2C inhibitors.

2) The effects of the mutations on the stability and function of 2C protein were not examined. 

Overall this manuscript is more suitable for a chemistry journal like the Journal of Medicinal Chemistry or ACS Pharmacology and Translational Research. 

Reviewer #3: The authors of this manuscript now submit a revised version that corrects the typographic errors in the first version, clarifies a number of points, and most importantly, demonstrates the antiviral properties of a new compound (12 b) resulting from the logical incorporation of properties of two derivatives of a molecule identified by another group (ref 20) via high-throughput screening. This new compound exhibits appreciable activity against representative strains from enterovirus species A, B, C, and D, and two of the 3 rhinovirus species. The paper is much improved by these changes. However, several straightforward changes should be made, in this reviewers opinion.

1. The authors state in their responses that "we agree that mentioning the therapeutically relevant range (for EC50 values) would improve the paper. We defined the therapeutically relevant window in line 793-796". However, lines 793 to 796 are in the section labeled "MEA recordings of spontaneous neuronal activity in rat primary cortical cultures". What is needed is a straight forward statement in the methods, such as saying that antiviral activity is defined as 50% inhibition of virus production or expression of luciferase at concentrations of 10 micromol/ L or less. Clearly submicromolar activity is typical of most antivirals in use, and it is exciting to see examples of low nanomolar range reported for compound 12b..

2. In the introduction, the authors state "Even though fluoxetine was well tolerated, it revealed no clinical benefit. Rather, the data pointed to a worsening of the patient conditions in the fluoxetine-treated cohort.(26) This is paradoxical; if people became more ill with fluoxetine, how can we say that it was well tolerated? Worsening of disease is obviously an adverse event to an antimicrobial agent..

3. Line 635 states that mechanistically, "the energy produced by ATP hydrolysis induces a series of conformational changes which drive unwinding of RNA or DNA." A reference should be provided that best supports this precise mechanistic statement, as until recently the helicase activity was in question, as it had been inconsistently demonstrated.

4. Isn't it an overstatement to say declare in the Discussion that " several comments exhibited " pan-EV and -RV activity" line 581? Pan clearly means "all" but the authors demonstrate data from only 2 of ~ 160 rhinovirus types, one-EV-B strain, and one EV-C strain (Sabin 1). Granted, at least activity is being seen against strains from 6 of 7 species...but it is an overstatement to call this pan-EV activity, and a more modest claim is appropriate.

---

## [Editor Report · Decision Letter 3]

22 Sep 2020

Dear Dr van Kuppeveld,

On behalf of my colleagues and the Academic Editor, Tobias Bollenbach, I am pleased to inform you that we will be delighted to publish your Research Article in PLOS Biology. 

Early Version

PRESS 

Kind regards,

Alice Musson

Publishing Editor, 

PLOS Biology

on behalf of

Di Jiang, PhD,

Senior Editor

PLOS Biology